# Chemical Diversity of Soft Coral Steroids and Their Pharmacological Activities

**DOI:** 10.3390/md18120613

**Published:** 2020-12-02

**Authors:** Ekaterina V. Ermolenko, Andrey B. Imbs, Tatyana A. Gloriozova, Vladimir V. Poroikov, Tatyana V. Sikorskaya, Valery M. Dembitsky

**Affiliations:** 1A.V. Zhirmunsky National Scientific Center of Marine Biology, 17 Palchevsky Str., 690041 Vladivostok, Russia; ecrire_711@mail.ru (E.V.E.); andrey_imbs@hotmail.com (A.B.I.); Miss.tatyanna@yandex.ru (T.V.S.); 2Institute of Biomedical Chemistry, bldg. 8, 10 Pogodinskaya Str., 119121 Moscow, Russia; tatyana.gloriozova@ibmc.msk.ru (T.A.G.); vladimir.poroikov@ibmc.msk.ru (V.V.P.); 3Centre for Applied Research, Innovation and Entrepreneurship, Lethbridge College, 3000 College Drive South, Lethbridge, AB T1K 1L6, Canada

**Keywords:** soft corals, steroids, chemical diversity, antitumor, anti-inflammatory, anti-eczemic, anti-psoriatic, biological activity prediction, PASS

## Abstract

The review is devoted to the chemical diversity of steroids produced by soft corals and their determined and potential activities. There are about 200 steroids that belong to different types of steroids such as secosteroids, spirosteroids, epoxy- and peroxy-steroids, steroid glycosides, halogenated steroids, polyoxygenated steroids and steroids containing sulfur or nitrogen heteroatoms. Of greatest interest is the pharmacological activity of these steroids. More than 40 steroids exhibit antitumor and related activity with a confidence level of over 90 percent. A group of 32 steroids shows anti-hypercholesterolemic activity with over 90 percent confidence. Ten steroids exhibit anti-inflammatory activity and 20 steroids can be classified as respiratory analeptic drugs. Several steroids exhibit rather rare and very specific activities. Steroids exhibit anti-osteoporotic properties and can be used to treat osteoporosis, as well as have strong anti-eczemic and anti-psoriatic properties and antispasmodic properties. Thus, this review is probably the first and exclusive to present the known as well as the potential pharmacological activities of 200 marine steroids.

## 1. Introduction

Soft corals belong to the order of Alcyonacea (class Octocorallia, Anthozoa, Cnidaria), formerly known as gorgonians [1,2] living in all the oceans of the world, especially in the tropics and subtropics. In addition, photosynthetic corals are successfully cultivated in artificial conditions and are the subject of chemical and biomedical investigation [3,4]. Soft corals produce more than 5800 secondary metabolites [5], including rare and unusual fatty acids [6,7,8,9,10,11,12,13], terpenoids [14,15,16], quinones [17], alkaloids [18,19,20,21,22,23], glycosides [24,25,26,27,28,29,30,31] and steroids [18,32,33,34,35,36,37]. It is known that many secondary metabolites isolated from marine invertebrates including soft corals, exhibit anticancer and other pharmacological activities [38,39,40,41,42,43,44,45].

In this review, we will look at rare and unusual steroids isolated from soft corals belonging to the order of Alcyonacea. The biological activity of many steroids has not been determined and we present the pharmacological activities detected experimentally and predicted based on the structure-activity relationships using the PASS (*Prediction of Activity Spectra for Substances*) software. PASS estimates the probabilities of several thousand biological activities with an average accuracy of about 96%. Probability of belonging to the class of “actives” Pa is calculated for each activity, providing the assessment of the hidden pharmacological potential of the investigated soft coral steroids [42]. 

## 2. Steroids Derived from the Genus *Sinularia*

*Sinularia* is a specific group of soft octocorals, a genus belonging to the family Alcyoniidae (class Anthozoa, phylum Cnidaria) and it includes about 175 actual species, 47 of which have been described only in the last quarter of a century [46,47]. *Sinularia* species are the most abundant corals in the entire Indo-Pacific region, especially in shallow water and dominate reef substrates [47].

Nowadays, many species of coral of the genus *Sinularia* are well adapted and cultured for biological and medical research and they are also an excellent source of many biologically active metabolites, including diterpenoids, unusual steroids and triterpenoids [15,19,38,40,48,49].

Three steroids, ergosta-5,24(28)-diene-3β,4α-diol (1), 24(*S*),28-epoxyergost-5-ene-3β,4α-diol (2) and dissesterol (**3**) were found in the methanol extract of the Vietnamese soft coral *Sinularia nanolobata*. Compound (**2**) exhibited moderate cytotoxicity against acute leukemia (HL-60) cell line with IC_50_ value of 33.5 µM and a weak effect on the hepatoma cancer (HepG2) and colon adenocarcinoma (SW480) cell lines with IC_50_ values of 64.3 and 71.0 µM, respectively [50]. The structures of the steroids (**1**–**18**) are shown in Figure 1 and the potential biological activities are shown in Table 1.

Steroids, 8αH-3β,11-dihydroxy-5α,6α-expoxy-24-methylene-9,11-secocholestan-9-one (**4**) and crassarosteroside A (**5**) were obtained from *Sinularia granosa* and *S. crassa* soft coral extracts [51,52], respectively. Steroids (**3**), 3β-hydroxyergosta-5,24(28)-diene-7-one (**6**), ergosta-3β,5α,6β-triol (**7**) have been isolated from *Sinularia conferta* and *S. nanolobata* [53,54] and compound (**7**) and 3β,7α-dihydroxyergosta-5,24(28)-diene (**8**) were detected in MeOH extract of the soft coral *S. cruciata* [55].

The reef soft coral *S. brassica*, which was cultured in an aquarium, afforded four steroids with methyl ester groups, sinubrasones A (**9**), B (**10**), D (**11**) and C (**12**). Compounds **9** and **10** were shown to exhibit significant cytotoxicity and compounds **11** and **12** were demonstrating attracting anti-inflammatory activities [56]. Steroids, 5,6β-epoxy-gorgosterol (**13**) and leptosteroid (**14**) were isolated and structurally elucidated from the Vietnamese soft coral *Sinularia leptoclados*. Both compounds obtained showed significant cytotoxicity against HepG2 (IC_50_ = 21.1 µM) and colon adenocarcinoma (IC_50_ = 28.6 µM) cell lines [54].

The cytotoxic steroid, crassarosterol A (**15**), which was originally found in the soft coral *S. crassa* [57] was later found in *S. arborea* [58] and *S. flexibilis* [59]. Analysis of its biological activity has shown that it exhibits cytotoxicity toward K562 and MOLT-4 leukemia cancer cells. Originally a steroid, 24-methyl-5β-cholest-24(28)-en-1α,3β,5β-triol-6-one, which was named gibberoketosterol (**16**) was identified from the lipophilic extracts of a Taiwanese soft coral *S. gibberosa* by Ahmed and co-workers in 2003 [60] and showed that to exhibit moderate cytotoxicity against the growth of Hepa59T/VGH cancer cells and it was later discovered in *S. numerosa* [61]. A methanol extract of the soft coral *S. microspiculata* revealed two sterols, 7-oxo-gorgosterol (**17**) and 16α-hydroxy-sarcosterol (**18**) [62]. Soft coral from Southern Taiwan *S. leptoclados* yielded a secosterol (**19**) [63]. 

The soft coral *S. flexibilis* appears to be one of the few corals most studied and described in the scientific literature. The structures of the steroids (**19**–**36**) are shown in Figure 2 and the potential biological activities are shown in Table 2. Yu and co-workers have used this organism to study steroids. Thus, a series of steroids (**20**–**25**) was isolated from the methanolic extract and their structures were established as: 5α,8α-epidioxygorgosta-6-en-3β-ol (**20**), 5α,8α-epidioxygorgosta-6,9(11)-dien-3β-ol (**23**), 22α,28-epidioxycholesta-5,23(*E*)-dien-3β-ol (**25A**), its C-22 epimer (**25B**) and compound (**24**) [64]. *S. nanolobata* yielded a sterol, sarcophytosterol (**26**), along with steroids (**3**) and (**6**) [65]. An unusual steroid, 3-(1′,2′-ethandiol)-cholest-3β,5α,6α,11α-tetraol (**27**) and secosteroid (**28**) were isolated from the South China sea gorgonian *S. suberosa* [66]. Two secosteroids (**29**) and (**30**) has been determined in Australian soft coral *Sinularia* sp. [67,68]. C18-Oxygenated steroidal ketal (**31**) was found in a Pacific soft coral extract of the genus *Sinularia* [69]. The minor sterol, ergosta-5,24(28),25-trien-3β-ol (**32**), was isolated from the soft coral extract of the genus *Sinularia*, its structure was determined and it was synthesized [70] and soft coral *S. dissecta* yielded polyhydroxylated sterol (**33**) [71,72] and a similar sterol (**34**) detected in the *S. numerosa* [73].

Kobayashi and co-workers conducted studies of the soft coral *S. mayi, S. gibberosa*, *S. dissecta* and *Sinularia* sp. collected off the coast of Japan and identified a series of steroids (**35**–**42**) but the biological activity of the isolated steroids was not determined [74,75,76,77,78]. The structures of the steroids (**37**–**58**) are shown in Figure 3 and the potential biological activities are shown in Table 3. A 11α,12α-epoxy-steroid (**43**) was identified in *S. dissecta* [79], a 3β,6α,9β,19α-tetraol-steroid (**44**) was derived from *S. inexplicita* [80]. 

Octocoral *S. leptoclados* is a source of bioactive 9,11-secosteroids and steroid 3β,11-dihydroxy-9,11-secogorgost-5-en-9-one (**45**) showed the highest the γ-peroxisome proliferator-activated receptor (PPARγ) activity with an IC_50_ value of 8.3 µM for inhibiting human breast adenocarcinoma cell (MCF-7) growth. In addition, this steroid modulated the expression of various PPARγ-regulated downstream biomarkers including cyclin D1, cyclin-dependent kinase, B-cell lymphoma 2 (Bcl-2), p38 and extracellular-signal-regulated kinase [81,82] and two cytotoxic 5,8-epidioxy-steroid (**46**) and also (**47**) were found in *Sinularia* sp. [83,84]. Similar steroidal glycosides (**48**–**51**) were isolated from water-methanol solutions of the soft coral *S. crispa* [85], *Sinularia* sp. [86] and *S. gibberosa* [87].

Two cytotoxic secosteroids, 22α-Acetoxy-24-methylene-3β,6α,11-trihydroxy-9,11-seco-cholest-7-en-9-one (**52**) and 11-acetoxy-24-methylene-1β,3β,6α-trihydroxy-9,11-seco-cholest-7-en-9-one (**53**) have been isolated from the soft coral *S. nanolobata* [61].

Two unusual steroidal derivatives named erectsterates A (**54**) and B (**55**), a pair of epimers at C-10, were isolated from the South China Sea soft coral *S. erecta*. Both compounds are rare steroids with a high degradation in ring B and an ester linkage between A and C/D rings, like the known compounds chaxines B and D from an edible mushroom *Agrocybe chaxingu* [88]. Compound (**55**) showed cytotoxic activity against A549 (human adenocarcinoma), HT-29 (human colorectal adenocarcinoma), SNU-398 (hepatocellular carcinoma) and Capan-1 (human pancreatic ductal adenocarcinoma) cancer cell lines [89].

A steroid named dissectolide (**56**) was purified from the methanol extract of soft coral *S. dissecta* [90]. The isolated steroid inhibits the growth of *Balanus amphitrite* larvae [91]. The ethyl acetate extract of a reef soft coral *S. brassica*, which was cultured in a tank, afforded two steroids, sinubrasones A (**57**) and B (**58**). Both highlighted products to exhibit significant cytotoxicity [92].

## 3. Steroids Derived from the other Coral’s Genera

The 17β, 20β-epoxy-23,24-dimethylcholest-5-ene-3β, 22-diacetate (**59**) and three unusual steroids (**60**–**62**) including secosteroid (**62**) were isolated from the Indian Ocean soft coral *Sarcophyton crassocaula* [93]. Cytotoxic steroids (**63**–**67**) were obtained from the acetone and MeOH extract of the soft coral *Nephthea erecta* [94]. A steroid with a spiro-ring A, B system named chabrolosteroid C (**66**) and chabrolosteroid A (**68**) were isolated from an organic extract of a Taiwanese soft coral *Nephthea chabrolii* [95]. The soft coral *Umbellulifera petasites* produces the steroid (**68**) and petasiterone B (**69**), as well as 5α-pregna-20-en-3-one (**70**), 5α,8α-epidioxycholest-6-en-3β-ol (**71**), 5α,8α-epidioxy-24(*S*)-methylcholesta-6,22-dien-3β-ol (**72**) and 5α,8α-epidioxy-24α-ethyl-cholesta-6,22-dien-3β-ol (**73**) have been found in the soft coral *Alcyonium gracillimum* [96,97,98]. The structures of the steroids (**59**–**84**) are shown in Figure 4 and the potential biological activities are shown in Table 4.

Unique highly oxygenated 13,17-secosteroids with split D ring were obtained from extracts of a Japanese octocoral of the genus *Dendronephthya* collected off the Izu Peninsula and named isogosterones A–D (**74**–**77**). The resulting steroids have inhibited the settlement of the *B. amphitrite* cyprid larvae [99]. Steroid, named 6-epi-yonarasterol B (**78**) was found in the Formosan gorgonian coral *Echinomuricea* sp. (family Plexauridae) [100].

Lactonic steroid derivatives with an unprecedented 1,10-secoergostane skeleton, stoloniolide I and II (**79** and **80**) and stoloniferones A (**81**), D (**82**), J (**83**), L (**84**) and O (**85**), which showed cytotoxic activity were found in the Okinawan soft coral *Clavularia viridis* [101,102,103,104]. 

Polyhydroxygenated steroids, hipposterone M (**86**), hipposterol G (**87**) and hippuristeroketal A (**88**) that demonstrated cytotoxicity against the anti-HCMV (human cytomegalovirus) were obtained from extracts of Taiwanese octocoral *Isis hippuris* collected at Orchid Island [105]. The structures of the steroids (**85**–**111**) are shown in Figure 5 and the potential biological activities are shown in Table 5 and Table 6. Steroid, named erectasteroid H (**89**), showed cytotoxic activity against P-388 (leukemia) and HT-29 [106] and spirosteroid (**90**) have been isolated from the Formosan soft coral *N. erecta* [107,108].

The coral *Klyxum flaccidum* produced the secosteroid, klyflaccisteroid K (**91**), which showed significant anti-inflammatory activity in suppressing superoxide anion generation and elastase release, with IC_50_ values of 5.8 and 1.5 µM, respectively [109] and the coral *Pinnigorgia* sp. produces bioactive 9,11-secosteroids, pinnisteroids A and C (**92** and **93**) displayed remarkable inhibitory effects on the generation of superoxide anions and the release of elastase in human neutrophils, with IC_50_ values from 2.3 to 3.3 µM [110]. The soft coral *Lobophytum laevigatum* contained unusual steroid, (22*S*,24*S*)-24-methyl-22,25-epoxyfurost-5-ene-3β,20β-diol (**94**) and demonstrated significantly upregulated PPAR transcriptional activity dose-dependently in Hep-G2 cells [111].

The 9,11-secosteroids, pinnigorgiols A (**95**), B (**96**) and E (**97**) with a rare carbon skeleton, a tricyclo[5,2,1]decane ring, were isolated from a gorgonian coral identified as *Pinnigorgia* sp. Isolated compounds displayed inhibitory effects on the generation of superoxide anions and the release of elastase by human neutrophils [112,113]. 16,22-Epoxy-20β,23S-dihydroxycholest-1-ene-3-one (**98**) unusual cholestane derivatives, was isolated from the South China Sea gorgonian coral *Subergorgia suberosa*. 

A series of cytotoxic steroids, verrucorosteroids A (**99**), B (**100**), D (**101**) and F (**102**), which demonstrated anticancer activity against eight human cancer cell lines as LNCaP (prostate cancer), HepG2, KB (epidermoid carcinoma), MCF-7, SK-Mel2 (melanoma), HL-60, LU-1 (lung cancer) and SW480 were isolated from the Vietnamese gorgonian *Verrucella corona* [114].

A series of steroids (**103**–**111**) was found and isolated for studying their biological activity from the genus *Alcyonium*. A steroid isolated from the Formosan soft coral *Alcyonium* sp. (Alcyoniidae) 3α,7α,12α-triacetoxy-5β-cholanic acid (**103**) [115] and steroids (**104**–**107**) were obtained from the crude extract of *A. gracillimum* which exhibited moderate cytotoxicity (IC_50_ 22 µg/mL) and antiviral activity (IC_50_ 8 µg/mL) against P388 and HSV-I (human α-herpesvirus), respectively [116,117]. Steroid derivatives 3-methoxy-19-norpregna-1,3,5(10),20-tetraene (**108**), 3-(4-O-acetyl6-deoxy-β-galactopyranosyloxy)-19-norpregna-1,3,5(10),20-tetraene (**109**) were isolated from *A. gracillimum*, which was collected from the Gulf of Sagami, Japan [118] and cytotoxic 24-methylcholest-4(5),24(28)-dien-3β,6β-diol (**110**) has been isolated from *A. patagonicum*, which was collected from the South China Sea [119]. The acetone extract of *Alcyonium* sp., which was collected from Taketomijima, Okinawa, yielded steroid, 3′-O-acetyl-pregnedioside-A (**111**) [120].

Another series of cytotoxic steroids called stereonsteroids A (**112**), B (**113**), D (**114**), F (**115**) and G (**116**, activity is shown in Table 7) were isolated from the methylene chloride extract of the Formosan soft coral *Stereonephthya crystalliana*. The extract of this coral showed significant cytotoxicity against A549, HT-29 and P-388 cancer cells in vitro [121]. Another two cytotoxic named sclerosteroids D (**117**) and E (**118**) were found in the soft coral *Scleronephthya gracillimum* [122]. Pregnane derivative 4-hydroxymethyl-5β-pregnan-3, 20-dione (**119**) has been isolated from the South China Sea gorgonian *Subergorgia suberosa* [123]. 

Marine withanolides, paraminabeolides A (**120**), B (**121**), C (**122**), D (**123**) and F (**124**) and same compounds named minabeolides 1 (**125**) and 5 (**126**) were obtained from lipid extracts of the Formosan soft coral *Paraminabea acronocephala*. Two compounds (**121**) and (**125**) demonstrated cytotoxic toward Hep G2 cancer cells [124]. Twenty years earlier Minabeolides-1 (**125**) and -5 (**126**) as C28 steroidal lactones of the withanolide class have been isolated from a soft coral *Minabea* sp., collected in Truk Lagoon [125].

Secosteroid (**127**) with epoxide at C-5 and C-6 group from the Formosan soft coral *Cespitularia hypotentaculata* exhibited cytotoxicity against HT-29 cells [126]. The structures of the steroids (**112**–**132**) are shown in Figure 6 and the potential biological activities are shown in Table 5. Two steroids, 11-acetoxy-9,11-secosterols, pinnisterols E (**128**) and I (**129**) with a 1,4-quinone moiety, were discovered from the gorgonian coral *Pinnigorgia* sp. Both identified compounds reduced elastase enzyme release [127]. (22R,23S,24S)-Polyoxygenated steroid named hippuristerone A (**130**) has been isolated from a Taiwanese gorgonian *I. hippuris* [128,129]. A rare steroid derivative named griffinipregnone (**131**) has been isolated from the octocoral *Dendronephthya griffin* and showed anti-inflammatory activity [130]. An unusual hemiketal steroid, named cladiellin A (**132**) was isolated from the soft coral *Cladiella* sp. [131] and a similar steroid 23-keto-cladiellin A (**133**) was obtained from the monohydroxylated sterol fraction of soft coral *Chromonephthea braziliensis* [132]. The structures of the steroids (**133**–**156**) are shown in Figure 7 and the potential biological activities are shown in Table 8. An unusual pentacyclic hemiacetal sterol nephthoacetal (**134**) and acetyl derivative (**135**) were isolated from soft coral *Nephthea* sp. Compound (**134**) exhibited a significant inhibitory effect with EC_50_ value of 2.5 μg/mL, while having low toxicity with LC_50_ > 25.0 μg/mL. The in vitro cytotoxic activity of two compounds exhibited moderate cytotoxicity with IC_50_ values of 12 and 10 μg/mL, respectively [133]. 

Extract of the soft coral *Dendronephthya gigantea* demonstrated the antiproliferative effect against the proliferation of HL-60 human leukemia cells and MCF-7 human breast cancer cells. The steroid 12-hydroxy-16,17-dimethyl-pregn-4-ene-1,20-dione (**136**) was isolated from the coral sterol fraction [134].

Krempenes A (**137**) and B (**138**) are unprecedented pregnane-type steroids that have been isolated from the marine soft coral *Cladiella krempfi* [135]. Steroid (**137**) contains a very unusual structural motif, with a hexacyclic oxadithiino unit fused to the steroidal ring A.

A rare steroidal hydroperoxide, 13,14-seco-22-norergosta-4,24(28)-dien-19-hydro-peroxide-3-one (**139**) has been found in the diethyl ether fraction of the Red Sea soft coral, *Litophyton arboretum* [136]. Steroid glycoside, dimorphoside B (**140**) has been isolated from the Western Pacific gorgonian *Anthoplexaura dimorpha* as the cell-division inhibitors in the sea urchin egg assay [137]. A unique unprecedented spinaceamine-bearing pregnane named scleronine (**141**) produces a Chinese soft coral *Scleronephthya* sp. [138]. 

Two secosteroids, 3β,11-dihydroxy-5β, 6β-epoxy-9,11-secocholestan-9-one (**142**) and 3β, 11-dihydroxy-5β,6β-epoxy-9,11-secogorgostan-9-one (**143**) have been found and identified from extracts of the Taiwanese soft coral *Cespitularia taeniata* [139,140,141].

Two steroids (**144** and **145**) and pregna-1,4,20-trien-3-one (**146**) have been isolated from the Pacific octocoral *Carijoa multiflora*. Compound (**144**) possesses a spiropregnane-based steroidal skeleton and showed antibacterial activity [142]. Compounds (**145** and **146**) and similar pregnane steroids (**147** and **148**) have been isolated from a gorgonian *Carijoa* sp. collected from the South China Sea. Compounds (**146**, **147** and **148**) exhibited cytotoxicity against the human hepatoma cell line Bel-7402, with IC_50_ values of 9.3, 11.0 and 18.6 µM, respectively [143]. The Hainan soft coral *Scleronephthya gracillimum* releases pregnane analogue (**149**) [144] and two unique chloro-pregnane steroids (**150** and **151**) have been isolated from the eastern Pacific octocoral *Carijoa multiflora* [145]. Another three chlorinated marine steroids, yonarasterols G (**152**), H (**153**) and I (**154**), were isolated from the Okinawan soft coral, *Clavularia viridis* [146]. 

Unusual steroid thioesters, parathiosteroids A (**155**) and C (**156**) were isolated from the 2-propanol extract of the soft coral *Paragorgia* sp. collected in Madagascar. Both compounds displayed cytotoxicity against a panel of three human tumor cell lines at the micromolar level [147].

The soft coral *Lobophytum michaelae* that lives on the coast of Taitung and its ethyl acetate extract contained three cytotoxic polyoxygenated steroids called michosterols A-C (**157**–**159**) [148] and also ethyl acetate extract the gorgonian *Leptogorgia* sp. collected from the South China Sea contained dihydroxy-ketosteroid (**160**) [149]. The structures of the steroids (**157**–**177**) are shown in Figure 8 and the potential biological activities are shown in Table 9.

The natural analog of calcitriol (1,25-dihydroxy-vitamin D3, **161**) called astrogorgiadiol (**162**) was first isolated in 1989 by Fusetani and co-workers [150] and another 9,10-secosteroids called calicoferols B (**163**), D (**164**) and E (**165**) have been isolated from an undescribed gorgonian of the genus *Muricella* [151,152]. 

Cytotoxic steroidal saponins, astrogorgiosides A (**166**) bearing acetamido-glucose moiety and astrogorgioside C (**167**) with a 19-nor and bearing an aromatized B ring steroid aglycone were obtained from the gorgonian *Astrogorgia dumbea* collected near Dongshan Island in the East China Sea. Both isolated compounds exhibited moderate cytotoxic activity with IC_50_ values of 26.8 and 45.6 µM against human tumor cells Bel-7402 and K562, respectively [153]. Rare pregnane derivative, 3-dimethoxy-5α-pregnan-20-one (**168**) has been detected in ethanol extract the Indian ocean gorgonian *Subergorgia suberosa*, collected from the Mandapam area, Tamil Nadu, India [154].

The cholestane class steroidal hemiacetals named anastomosacetals A (**169**) and D (170) were obtained from the gorgonian coral *Euplexaura anastomosans*, collected off the shore of Keomun Island, South Sea Korea [155]. Gorgonian species, *Bebryce indica*, collected off the coast of Sanya (Hainan, China) was found to contain steroidal glycoside named bebrycoside (**171**) [156]. Bebrycoside (**171**) and similar 27-O-[β-D-arabino-pyranosyloxy]-20β,22α-dihydroxy-cholest-4-ene-3-one, named muricellasteroid D (**172**) and also rare steroid 22α-O-acetyl-2β-O-methylene-[4β-hydroxy-phenly]-cholest-4-ene-3-one, named muricellasteroid E (**173**) have been isolated from the EtOH/CH_2_Cl_2_ extracts of the South China Sea gorgonian coral *Muricella flexuosa*. Compounds **172** and **173** showed moderate cytotoxicity against A375, K562 and A549 cancer cell lines [157].

Rare, a spiroketal steroid, 22-acetoxy-3,25-dihydroxy-16,24,20-24-bisepoxy-(3β,16α,20*S*,22*R*, 24*S*)-cholest-5-ene (**174**) was isolated from the Indian Ocean gorgonian, *Gorgonella umbraculum* [158] and trihydroxy sterol, pregna-5-ene-3β,20α,21-triol (**175**) has been isolated from the Gulf of California gorgonian *Muricea* cf. *austera* [159]. Oxysterols (**176**–**181**) were detected and identified from an octocoral of the genus Gorgonia from the eastern Pacific of Panama [160]. The structures of the steroids (**178**–**187**) are shown in Figure 9 and the potential biological activities are shown in Table 10.

The *I. hippuris*, as well as other species of this genus, belong to the bamboo corals that live in the Central-West Pacific, the Indian Ocean, as well as in the Red Sea [161]. Studies of these corals in the last quarter of a century have shown that they satisfy their metabolic requirements for carbon through the products of photosynthesis [162] and only about 9% from bacterioplankton. These corals synthesize many metabolites, which are highly oxidized products that are oxidized by their own endogenous oxygen. These include numerous highly oxygenated and spiroketal steroids, many of which demonstrate anticancer activity against many cancer cells [163,164,165,166,167,168,169,170].

Over twenty polyoxygenated steroids, including (22S)-2α,3α-diacetoxy-11β,18β-dihydroxy-24-methyl-18,20β,22,25-diepoxy-5α-furostane (**182**), (22S)-2α,3α-diacetoxy-11β-hydroxy-24-methyl-22,25-epoxy-5α-furostan-18,20β-lactone (**183**) as well as (**184**–**187**) were extracted with a mixture of n-hexane and CH_2_Cl_2_ from the *I. hippuris* living in the southeast coast of Taiwan. Many of the steroids showed cytotoxic activity against Hep G2, Hep 3B, A549, MCF-7 and MDAMB-231 cells [171].

## 4. Comparison of Biological Activities of Natural Soft Coral Steroids

The biological activity of the molecule depends on its structure, which allows analyzing the structure-activity relationships (SAR). This idea was first proposed by Brown and Fraser in 1868 [172] more than 150 years ago; however, it was further developed in the mid-1970s [173,174]. 

The quantitative structure-activity relationships (QSAR) paradigm was first implemented in toxicology, pharmaceutical and medicinal chemistry, and, ultimately, various aspects of organic and bioorganic chemistry [175]. For over 50 years, the QSAR paradigm has been widely used due to its original postulate that activity was a function of the structure described by electronic attributes, hydrophobicity or steric properties [176]. The rapid and extensive development of methodologies and computational methods led to the definition and refinement of many approaches that introduce the paradigm into the practice of research and development [177].

Several computer programs can estimate with some degree of certainty the pharmacological activities of organic metabolites isolated from natural sources or synthetic compounds [178,179,180]. Classical (Q)SAR methods are based on the analysis of (quantitative) structure-activity relationships for a single or several biological activities using the compounds belonging to the same chemical series as the training set [181]. 

Computer program PASS, which is continuously updating and improving for the past thirty years [182], is based on the analysis of a heterogeneous training set included information about more than a million known biologically active compounds with data on ca. 10,000 biological activities [183,184]. Chemical descriptors implemented in PASS, which reflect the peculiarities of ligand-target interactions and original realization of the Bayesian approach for elucidation of structure-activity relationships provide the average accuracy and predictivity for several thousand biological activities equal to about 96% [185,186]. In several comparative studies, it was shown that PASS outperforms in predictivity some other recently developed methods for estimation of biological activity profiles [187,188]. Freely available via Internet PASS Online web-service [189] is used by more than twenty thousand researchers from almost a hundred countries to determine the most promising biological activities for both natural and synthetic compounds [184,186,190,191,192,193]. To reveal the hidden pharmacological potential of the natural substances, we are successfully using PASS for the past fifteen years [194,195,196,197,198].

In the current study, we obtained PASS predictions for two hundred steroids produced by soft corals. PASS estimates are presented as Pa values, which correspond to the probability of belonging to a particular class of “*actives*” for each predicted biological activity. The higher the Pa value is, the higher is the confidence that the experiment will confirm the predicted biological activity [186].

We have selected about 200 steroids, of which 50 belong to the genus *Sinularia*, which represent different types of steroids such as secosteroids, spirosteroids, epoxy- and peroxy-steroids, steroid glycosides, halogenated steroids and steroids containing sulfur or nitrogen heteroatoms. The types of steroids presented represent the chemical diversity of these secondary metabolites. Therefore, the pharmacological activities of various types of steroids are of great interest.

Analyzing the data obtained using PASS, we can state that almost all steroids presented in this article exhibit potential anti-tumor activity with varying degrees of reliability. In addition, forty-one steroids demonstrate anti-tumor and related activity with a confidence level of more than 90 percent, which is of significant interest to the pharmaceutical industry. Figure 10 shows the distribution of steroids with antitumor and related activities in the corals of the genus *Sinularia*. This group of steroids has a high degree of certainty over 90 percent.

We highlighted other activities in a separate column and named them *Lipid metabolism regulators*, which include such properties of steroids as anti-hypercholesterolemic, treatment of atherosclerosis, cholesterol synthesis inhibitor and hypolipemic activity. This group includes 32 steroids that show anti-hypercholesterolemic activity with over 90 percent confidence. It is known that hypercholesterolemia and oncogenesis are interrelated, as shown in many studies [199,200,201,202,203]. Therefore, these data are also of great practical interest in identifying the etiology of cancer and its treatment. The column additional activity presents the activities that steroids demonstrate and some activities can also be attributed to the main ones since the reliability of the activities of some steroids exceeds 90 percent of the reliability. For example, the crassarosteroside A (**5**) water-soluble steroidal glycoside isolated from the Soft Coral *S. crassa*, according to the authors of the article, demonstrated cytotoxicity against human liver carcinoma (HepG2 and HepG3) [51,52]. Our studies have shown that it demonstrates strong antitumor activity and can be successfully used for the treatment of proliferative diseases, in addition, it demonstrates anti-hypercholesterolemic activity and is also a respiratory analeptic. Figure 11 shows the predicted and calculated (log activities) pharmacological activities of crassarosteroside A.

In addition, some coral steroids show interesting pharmacological activities with a high degree of confidence, over 90%. For example, the following steroids **15**, **32**, **76**, **138**, **149**, **150**, **184**, **185** and **187** show anti-inflammatory activity. And steroids **5**, **14**, **15**, **50**, **51**, **65**, **101**, **114**, **115**, **116**, **155**, **169**, **175**, **176**, **177**, **178**, **180** and **184** can be classified as respiratory analeptic drugs, which are central nervous system stimulants.

Steroids have also been found that exhibit rather rare and extremely specific activities. For example, steroids **18**, **37** and **38** can be used to treat autoimmune diseases. Steroids **53**, **54** and **161** show anti-osteoporotic properties and can be used to treat osteoporosis. Steroids **94** and **102** are immunomodulators that can be used in the treatment of patients with AIDS. Steroid **163** exhibited strong anti-eczematic and anti-psoriatic properties and steroid **125** exhibited anti-eczematic and spasmolytic properties. Steroid **142** is a hepato-protector, steroid **86** is an inhibitor of angiogenesis and steroid **87** can be used as a general anesthetic.

## 5. Conclusions

About 200 soft coral steroids are classified as different types of steroids such as secosteroids, spirosteroids, epoxy- and peroxy-steroids, steroid glycosides, halogenated steroids, polyoxygenated steroids and steroids containing sulfur or nitrogen heteroatoms. There also have been found steroids that exhibit rather rare and extremely specific activities. The PASS program is constantly evolving by increasing the database of both natural and synthetic compounds and increasing biological activities by incorporating experimental data. Currently, PASS contains over 1,000,000 chemical structures of natural and synthetic compounds associated with over 10,000 biological activities. These activities are included in the program and are taken from published articles, reviews and other official medical profile documents. Thus, analyzing the presented steroids, it can be stated that more than 40 steroids demonstrate antitumor and related activity with a confidence level of more than 90 percent. Another group, which consists of 32 steroids, demonstrates anti-hypercholesterolemic activity with more than 90 percent confidence.

In addition, ten steroids exhibit anti-inflammatory activity and 20 steroids can be classified as respiratory analeptic drugs. Some steroids with rare structures exhibit anti-osteoporotic properties and can be used to treat osteoporosis, as well as exhibit strong anti-eczemic and anti-psoriatic properties, exhibit anti-eczema and antispasmodic properties. Thus, these data show that soft coral steroids are very interesting in terms of their medical use. However, this requires additional extensive research.

## Figures and Tables

**Figure 1 marinedrugs-18-00613-f001:**
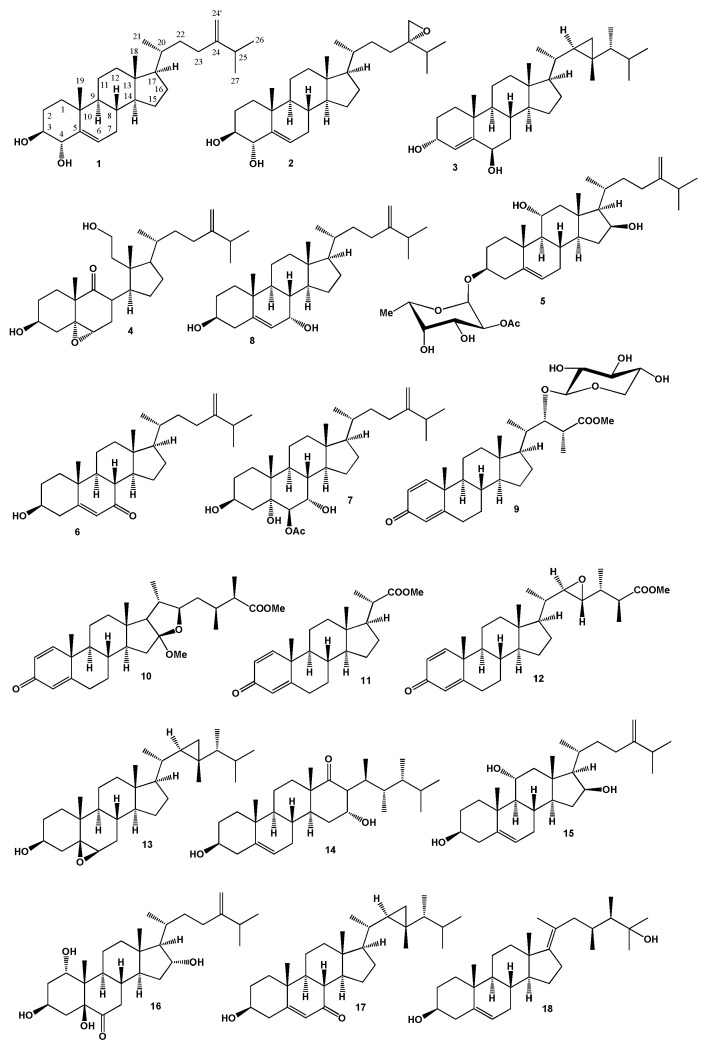
Bioactive steroids derived from the genus *Sinularia*.

**Figure 2 marinedrugs-18-00613-f002:**
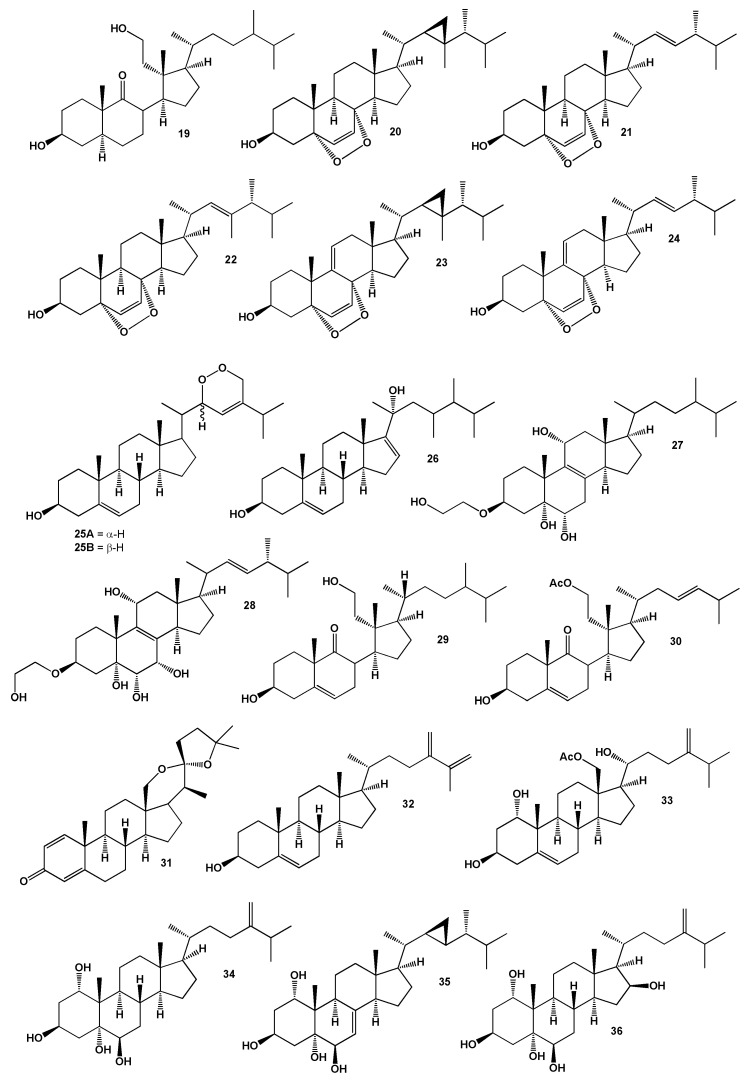
Bioactive steroids derived from the genus *Sinularia*.

**Figure 3 marinedrugs-18-00613-f003:**
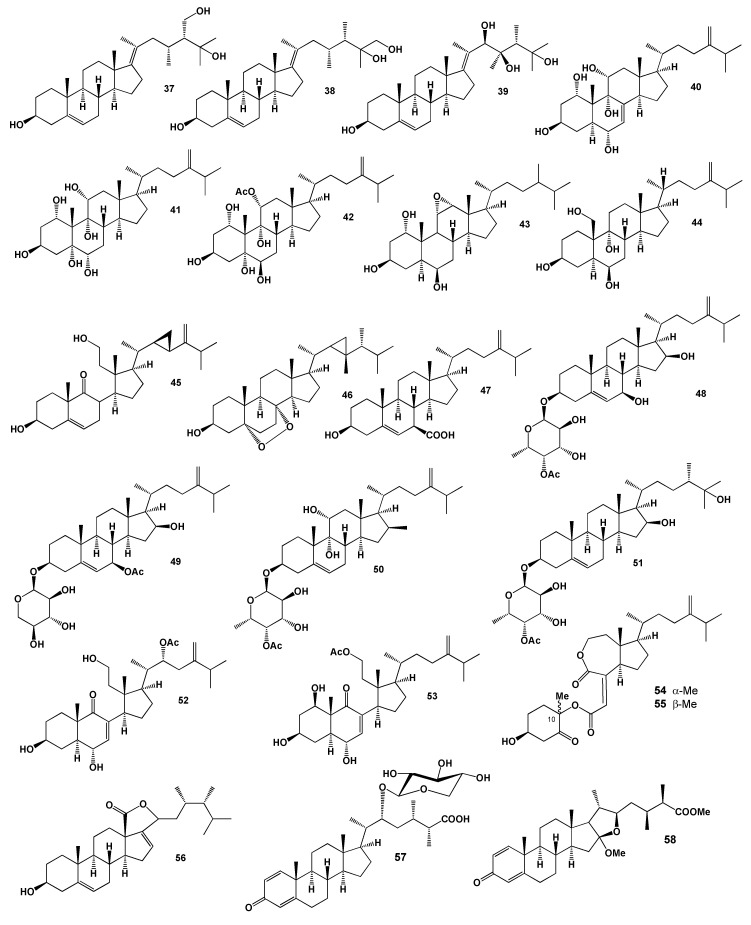
Bioactive steroids derived from the genus *Sinularia*.

**Figure 4 marinedrugs-18-00613-f004:**
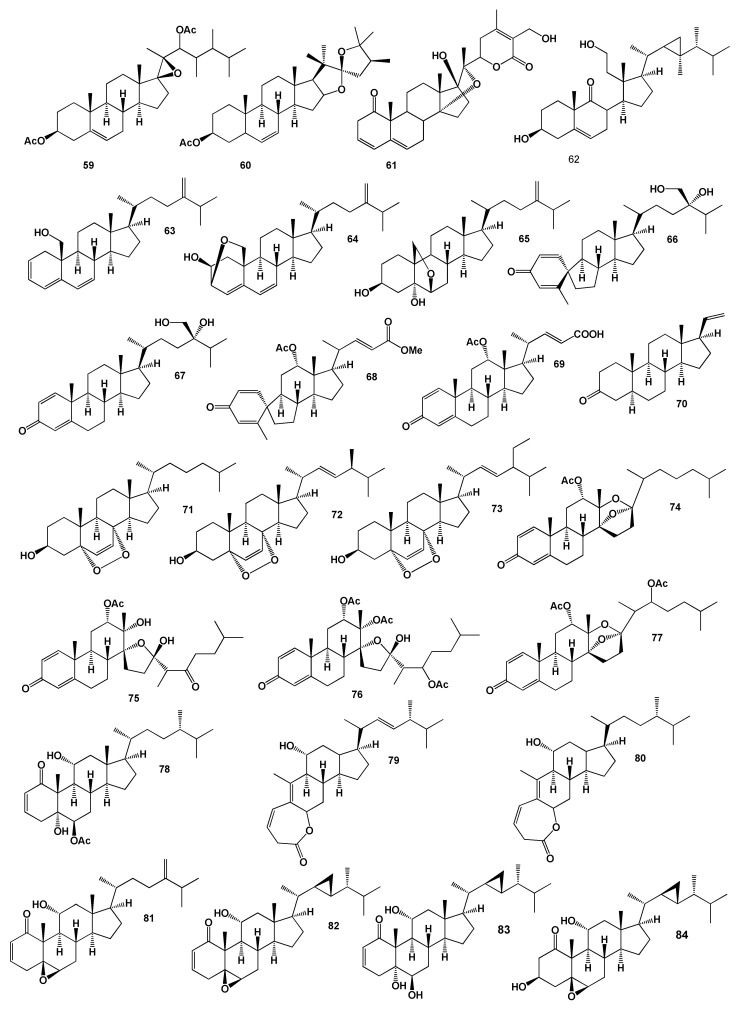
Bioactive steroids derived from soft corals.

**Figure 5 marinedrugs-18-00613-f005:**
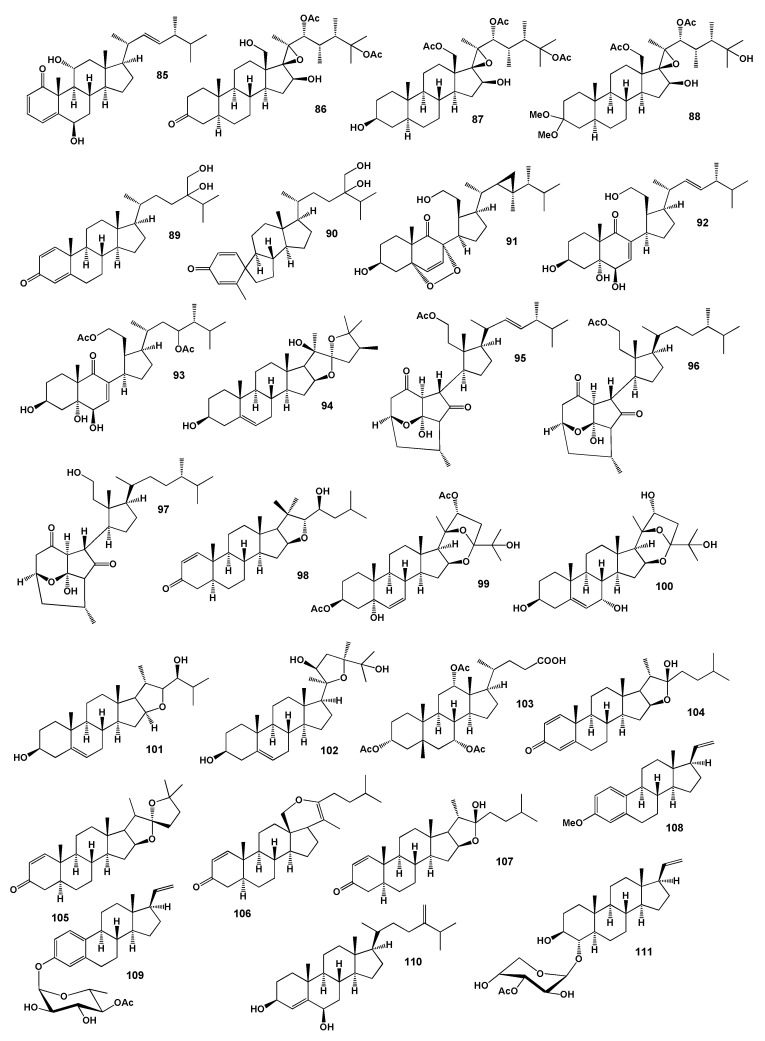
Bioactive steroids derived from soft corals.

**Figure 6 marinedrugs-18-00613-f006:**
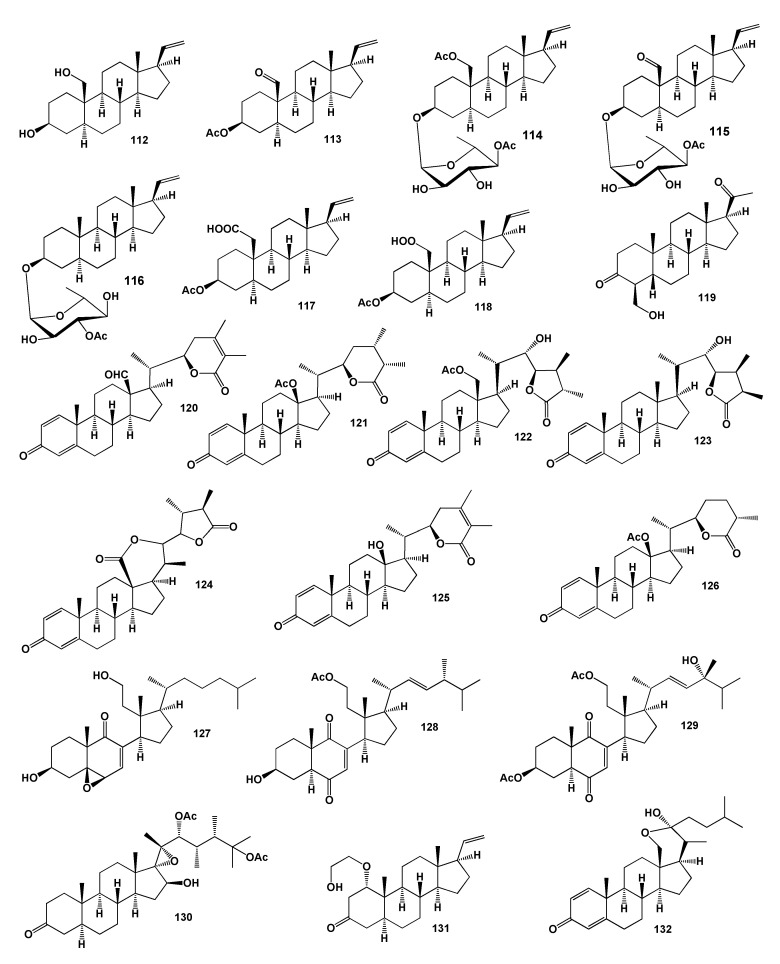
Bioactive steroids derived from soft corals.

**Figure 7 marinedrugs-18-00613-f007:**
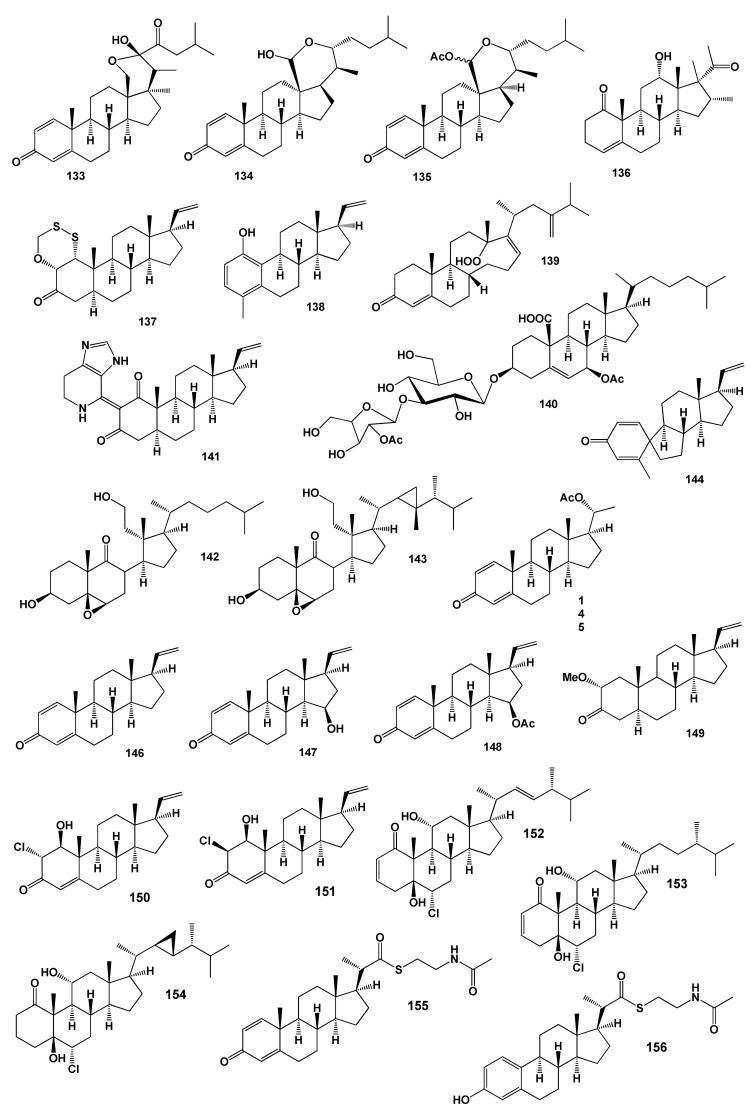
Bioactive steroids derived from soft corals.

**Figure 8 marinedrugs-18-00613-f008:**
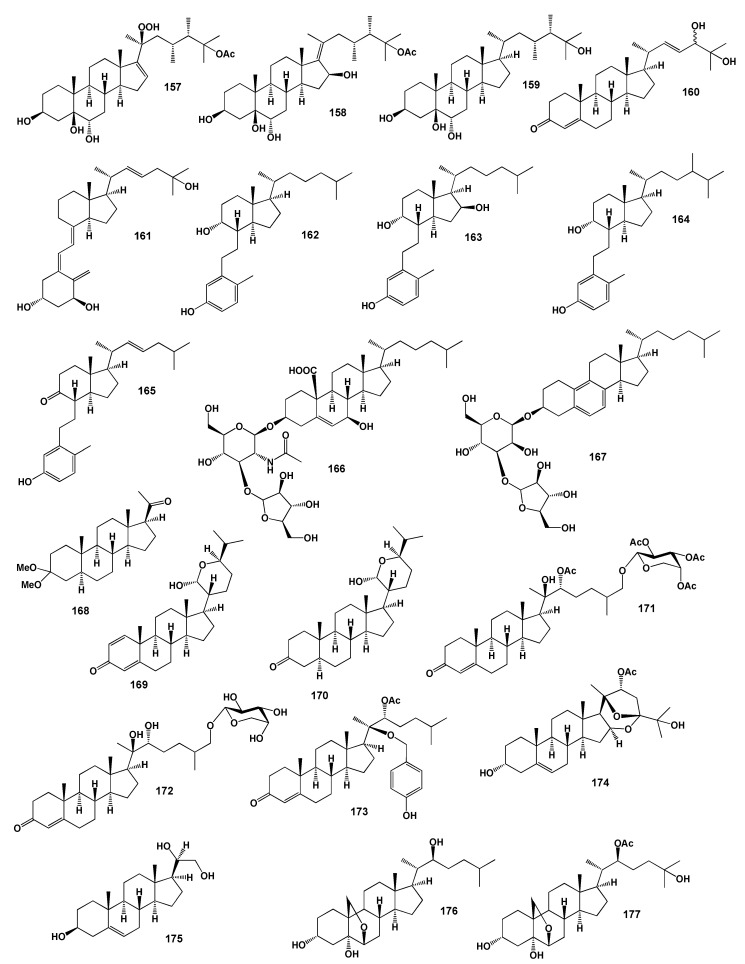
Bioactive steroids derived from soft corals.

**Figure 9 marinedrugs-18-00613-f009:**
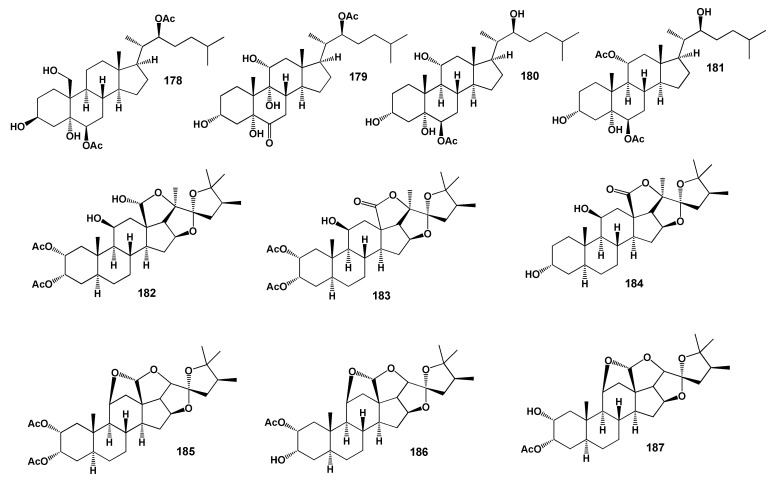
Bioactive steroids derived from soft corals.

**Figure 10 marinedrugs-18-00613-f010:**
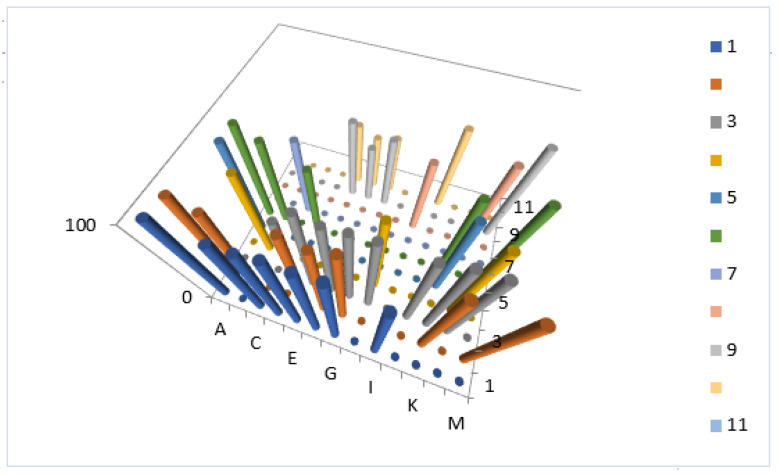
3D column graph of steroids derived from soft corals of the genus *Sinularia* that show antitumor and related activities. The letters represent the steroid numbers shown in Figure 1, Figure 2 and Figure 3 and Table 1, Table 2 and Table 3: A—(**5**), B—(**9**), C—(**18**), D—(**20**), E—(**21**), F—(**22**), G—(**24**), H—(**32**), I—(**37**), J—(**39**), K—(**49**), L—(**50**) and M—(**51**). Steroids that belong to this group, according to the data obtained by the PASS, have confirmed more than 90 percent of their biological activity.

**Figure 11 marinedrugs-18-00613-f011:**
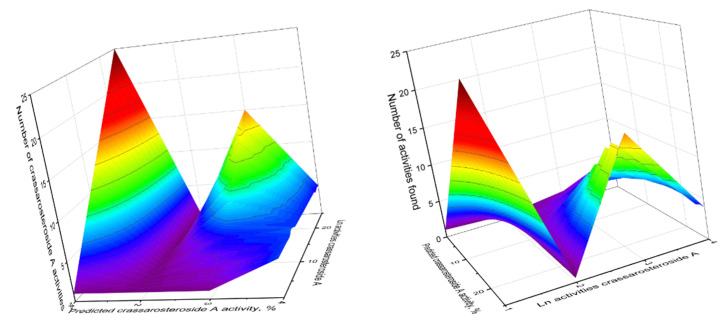
3D Graph (X and Y views, predicted and log calculated activities) of the pharmacological activities of water-soluble 3-O-[2′-O-acetyl-(α-L-fuco-pryranosyl)]-11,16-hydroxy-24-methylene-cholesterol which called crassarosteroside A (**5**) was isolated from *Sinularia granosa* and *S. crassa* soft coral extracts. According to PASS data, this glycoside demonstrated 25 different activities, with eight activities having a found confidence of more than 90 percent. The main pharmacological activities of crassarosteroside A (**5**) are respiratory analeptic (99.2%), proliferative diseases treatment (96.7%), chemopreventive (95.8%), anti-hypercholesterolemic (94.7%), hepatoprotectant (94.5%), anticarcinogenic (90.7%), antineoplastic (90.2%) and anti-inflammatory (90.0%, Table 1).

**Table 1 marinedrugs-18-00613-t001:** Probable biological activities of steroids derived from the genus *Sinularia* estimated by PASS (*Prediction of Activity Spectra for Substances*).

No.	Antitumor and Related Activity, (Pa) *	Lipid Metabolism Regulators, (Pa) *	Additional Biological Activity, (Pa) *
**1**	Antineoplastic (0.840) Chemopreventive (0.806) Proliferative diseases treatment (0.705) Antimetastatic (0.562) Anticarcinogenic (0.542)	Anti-hypercholesterolemic (0.920) Hypolipemic (0.798) Atherosclerosis treatment (0.637) Hyperparathyroidism treatment (0.523)	Respiratory analeptic (0.879) Anesthetic general (0.851) Antifungal (0.792) Anti-inflammatory (0.754) Antibacterial (0.545)
**2**	Antineoplastic (0.833) Chemopreventive (0.738) Proliferative diseases treatment (0.643) Antimetastatic (0.626)	Anti-hypercholesterolemic (0.840) Hypolipemic (0.646) Atherosclerosis treatment (0.634)	Anti-inflammatory (0.806) Anesthetic general (0.800) Respiratory analeptic (0.789) Antifungal (0.616)
**3**	Antineoplastic (0.765) Dementia treatment (0.512)	Anti-hypercholesterolemic (0.655) Hypolipemic (0.530)	Immunosuppressant (0.735) Anti-inflammatory (0.667)
**4**	Antineoplastic (0.798) Apoptosis agonist (0.751) Antimetastatic (0.524)	Hypolipemic (0.732) Cholesterol synthesis inhibitor (0.545) Antileukemic (0.539)	Immunosuppressant (0.759) Antifungal (0.748) Antibacterial (0.642)
**5**	Proliferative diseases treatment (0.967) Chemopreventive (0.958) Anticarcinogenic (0.907) Antineoplastic (0.902) Dementia treatment (0.508)	Anti-hypercholesterolemic (0.947) Hypolipemic (0.781) Atherosclerosis treatment (0.609)	Respiratory analeptic (0.992) Anti-inflammatory (0.900) Antifungal (0.880) Neuroprotector (0.849) Antibacterial (0.797)
**6**	Chemopreventive (0.839) Antineoplastic (0.834) Proliferative diseases treatment (0.716) Antimetastatic (0.573)	Anti-hypercholesterolemic (0.937) Hypolipemic (0.809) Neuroprotector (0.767) Atherosclerosis treatment (0.592)	Respiratory analeptic (0.896) Anesthetic general (0.828) Anti-inflammatory (0.778) Antifungal (0.768)
**7**	Antineoplastic (0.861) Chemopreventive (0.827) Proliferative diseases treatment (0.779) Anticarcinogenic (0.565) Antimetastatic (0.560)	Anti-hypercholesterolemic (0.883) Hypolipemic (0.771) Atherosclerosis treatment (0.675)	Respiratory analeptic (0.868) Antifungal (0.821) Anti-inflammatory (0.755) Antibacterial (0.603) Antimycobacterial (0.569)
**8**	Antineoplastic (0.852) Apoptosis agonist (0.801) Proliferative diseases treatment (0.766) Anticarcinogenic (0.732)	Anti-hypercholesterolemic (0.955) Hypolipemic (0.878) Neuroprotector (0.788) Atherosclerosis treatment (0.658)	Respiratory analeptic (0.856) Antifungal (0.780) Anti-inflammatory (0.750) Antimycobacterial (0.653)
**9**	Chemopreventive (0.902) Antineoplastic (0.858) Proliferative diseases treatment (0.821)	Anti-hypercholesterolemic (0.911) Atherosclerosis treatment (0.548) Hypolipemic (0.548)	Anti-inflammatory (0.896) Antifungal (0.765)
**10**	Antineoplastic (0.776) Apoptosis agonist (0.601)		Antiprotozoal (Plasmodium) (0.778) Anti-inflammatory (0.765)
**11**	Antineoplastic (0.807) Apoptosis agonist (0.747) Proliferative diseases treatment (0.596)	Anti-hypercholesterolemic (0.630)	Anti-inflammatory (0.930) Anti-asthmatic (0.627)
**12**	Antineoplastic (0.838)		Anti-inflammatory (0.792)
**13**	Apoptosis agonist (0.790) Antineoplastic (0.781)		Respiratory analeptic (0.772) Immunosuppressant (0.744) Antifungal (0.708)
**14**	Antineoplastic (0.824) Proliferative diseases treatment (0.610)	Anti-hypercholesterolemic (0.909) Lipid metabolism regulator (0.672)	Respiratory analeptic (0.907) Anti-inflammatory (0.673)
**15**	Antineoplastic (0.865) Proliferative diseases treatment (0.812) Apoptosis agonist (0.767) Anticarcinogenic (0.730)	Anti-hypercholesterolemic (0.930) Hypolipemic (0.827) Atherosclerosis treatment (0.646)	Respiratory analeptic (0.970) Anti-inflammatory (0.904) Anesthetic general (0.850) Antibacterial (0.586)
**16**	Antineoplastic (0.838) Proliferative diseases treatment (0.729) Antimetastatic (0.651)	Anti-hypercholesterolemic (0.909) Hypolipemic (0.786)	Respiratory analeptic (0.937) Antifungal (0.789) Cardiotonic (0.626)
**17**	Antineoplastic (0.730)	Anti-hypercholesterolemic (0.679)	Immunosuppressant (0.722)
**18**	Antineoplastic (0.925) Proliferative diseases treatment (0.787) Prostate cancer treatment (0.670)	Anti-hypercholesterolemic (0.913) Hypolipemic (0.626) Lipid metabolism regulator (0.566)	Autoimmune disorders treatment (0.908) Respiratory analeptic (0.780) Antifungal (0.525)

* Pa is the probability of belonging to the class of “actives”; only activities with Pa > 0.5 are shown.

**Table 2 marinedrugs-18-00613-t002:** Probable biological activities of steroids derived from the genus *Sinularia* estimated by PASS.

No.	Antitumor and Related Activity, (Pa) *	Lipid Metabolism Regulators, (Pa) *	Additional Biological Activity, (Pa) *
**19**	Antineoplastic (0.759) Antimetastatic (0.661)	Anti-hypercholesterolemic (0.800) Hypolipemic (0.653)	Anesthetic general (0.768) Antibacterial (0.615)
**20**	Apoptosis agonist (0.941) Antineoplastic (0.761) Chemopreventive (0.687)	Atherosclerosis treatment (0.747) Lipoprotein disorders treatment (0.567)	Antiprotozoal (Plasmodium) (0.791) Antioxidant (0.650) Antifungal (0.634)
**21**	Apoptosis agonist (0.982) Chemopreventive (0.945) Antineoplastic (0.920) Prostate cancer treatment (0.680)	Atherosclerosis treatment (0.909) Hypolipemic (0.831) Lipoprotein disorders treatment (0.816) Anti-hypercholesterolemic (0.799)	Antiprotozoal (Plasmodium) (0.845) Immunosuppressant (0.733) Antiparkinsonian, rigidity relieving (0.730)
**22**	Apoptosis agonist (0.969) Antineoplastic (0.897) Chemopreventive (0.880) Antimetastatic (0.645) Proliferative diseases treatment (0.513)	Atherosclerosis treatment (0.841) Lipoprotein disorders treatment (0.701) Hypolipemic (0.645) Anti-hypercholesterolemic (0.559)	Antiprotozoal (Plasmodium) (0.828) Antioxidant (0.670) Immunosuppressant (0.653) Antifungal (0.619) Anti-inflammatory (0.523)
**23**	Apoptosis agonist (0.962) Chemopreventive (0.751) Antineoplastic (0.740)	Atherosclerosis treatment (0.626)	Antifungal (0.630) Antiprotozoal (Plasmodium) (0.623) Antileukemic (0.552)
**24**	Apoptosis agonist (0.994) Chemopreventive (0.966) Antineoplastic (0.905) Prostate cancer treatment (0.546)	Atherosclerosis treatment (0.833) Hypolipemic (0.785) Anti-hypercholesterolemic (0.746) Lipoprotein disorders treatment (0.594)	Antiprotozoal (Plasmodium) (0.693) Immunosuppressant (0.692) Antiparkinsonian, rigidity relieving (0.672)
**25A**	Antineoplastic (0.816) Proliferative diseases treatment (0.577)	Anti-hypercholesterolemic (0.788) Hypolipemic (0.655)	Antiprotozoal (Plasmodium) (0.868) Immunosuppressant (0.766)
**25B**	Antineoplastic (0.819) Proliferative diseases treatment (0.581)	Anti-hypercholesterolemic (0.784) Hypolipemic (0.635)	Antiprotozoal (Plasmodium) (0.861) Immunosuppressant (0.756)
**26**	Antineoplastic (0.758) Chemopreventive (0.757) Proliferative diseases treatment (0.642) Prostate cancer treatment (0.598)	Anti-hypercholesterolemic (0.867) Anxiolytic (0.736) Hypolipemic (0.640) Atherosclerosis treatment (0.526)	Respiratory analeptic (0.809) Immunosuppressant (0.711) Anti-inflammatory (0.689) Antifungal (0.560)
**27**	Antineoplastic (0.886) Apoptosis agonist (0.856)	Anti-hypercholesterolemic (0.686) Atherosclerosis treatment (0.630)	Antifungal (0.739) Anti-inflammatory (0.696)
**28**	Antineoplastic (0.876) Apoptosis agonist (0.874) Chemopreventive (0.696)	Hypolipemic (0.724) Anti-hypercholesterolemic (0.677) Atherosclerosis treatment (0.620)	Immunosuppressant (0.669) Anesthetic (0.570) Anti-inflammatory (0.553)
**29**	Antineoplastic (0.763) Chemopreventive (0.651) Proliferative diseases treatment (0.541)	Anti-hypercholesterolemic (0.831) Hypolipemic (0.625) Prostate disorders treatment (0.536)	Respiratory analeptic (0.625) Antibacterial (0.573) Antifungal (0.558)
**30**	Antineoplastic (0.807) Proliferative diseases treatment (0.651)	Anti-hypercholesterolemic (0.915) Lipid metabolism regulator (0.768)	Anti-inflammatory (0.736) Antibacterial (0.582)
**31**	Antineoplastic (0.885)	Hypolipemic (0.616)	Anti-inflammatory (0.911)
**32**	Antineoplastic (0.910) Proliferative diseases treatment (0.882)	Anti-hypercholesterolemic (0.955) Atherosclerosis treatment (0.596)	Respiratory analeptic (0.959) Antiprotozoal (Plasmodium) (0.667)
**33**	Antineoplastic (0.831) Proliferative diseases treatment (0.698) Apoptosis agonist (0.593)	Anti-hypercholesterolemic (0.903) Hypolipemic (0.685) Lipid metabolism regulator (0.546)	Anti-inflammatory (0.870) Antifungal (0.783) Neuroprotector (0.636)
**34**	Apoptosis agonist (0.864) Antineoplastic (0.860) Proliferative diseases treatment (0.700)	Anti-hypercholesterolemic (0.889) Atherosclerosis treatment (0.699) Lipoprotein disorders treatment (0.587)	Antifungal (0.769) Anti-inflammatory (0.735) Antibacterial (0.550)
**35**	Antineoplastic (0.805)	Hypolipemic (0.529)	Immunosuppressant (0.640)
**36**	Antineoplastic (0.883) Apoptosis agonist (0.848) Proliferative diseases treatment (0.655)	Hypolipemic (0.795) Anti-hypercholesterolemic (0.772) Atherosclerosis treatment (0.666)	Anti-inflammatory (0.763) Antifungal (0.753) Antibacterial (0.588)

* Only activities with Pa > 0.5 are shown.

**Table 3 marinedrugs-18-00613-t003:** Biological activities of steroids derived from the genus *Sinularia* estimated by PASS.

No.	Antitumor and Related Activity, (Pa) *	Lipid Metabolism Regulators, (Pa) *	Additional Biological Activity, (Pa) *
**37**	Antineoplastic (0.915) Proliferative diseases treatment (0.722)	Anti-hypercholesterolemic (0.910) Lipid metabolism regulator (0.675)	Autoimmune disorders treatment (0.889) Neuroprotector (0.700)
**38**	Antineoplastic (0.895) Proliferative diseases treatment (0.673)	Anti-hypercholesterolemic (0.910) Atherosclerosis treatment (0.695) Lipid metabolism regulator (0.575)	Autoimmune disorders treatment (0.874) Anti-inflammatory (0.717) Neuroprotector (0.700)
**39**	Antineoplastic (0.903) Proliferative diseases treatment (0.768) Dementia treatment (0.554)	Anti-hypercholesterolemic (0.947) Lipid metabolism regulator (0.768) Hypolipemic (0.713)	Autoimmune disorders treatment (0.829) Neuroprotector (0.835) Anti-inflammatory (0.786)
**40**	Apoptosis agonist (0.823) Antineoplastic (0.815) Proliferative diseases treatment (0.610)	Hypolipemic (0.901) Anti-hypercholesterolemic (0.754) Atherosclerosis treatment (0.582)	Antifungal (0.737) Anti-inflammatory (0.705) Antibacterial (0.597)
**41**	Apoptosis agonist (0.845) Antineoplastic (0.823) Proliferative diseases treatment (0.678)	Hypolipemic (0.790) Anti-hypercholesterolemic (0.687) Atherosclerosis treatment (0.659)	Anti-fungal (0.761) Anti-inflammatory (0.735)
**42**	Apoptosis agonist (0.851) Antineoplastic (0.824) Proliferative diseases treatment (0.739)	Hypolipemic (0.769) Anti-hypercholesterolemic (0.733) Atherosclerosis treatment (0.631)	Antifungal (0.783) Anti-inflammatory (0.775) Antibacterial (0.543)
**43**	Antineoplastic (0.890) Apoptosis agonist (0.743) Proliferative diseases treatment (0.551)	Anti-hypercholesterolemic (0.793) Hypolipemic (0.767) Atherosclerosis treatment (0.638)	Anti-inflammatory (0.728) Antifungal (0.725) Immunosuppressant (0.704)
**44**	Antineoplastic (0.841) Apoptosis agonist (0.770) Proliferative diseases treatment (0.760)	Anti-hypercholesterolemic (0.900) Hypolipemic (0.843) Atherosclerosis treatment (0.620)	Cardiotonic (0.872) Anesthetic general (0.847) Antifungal (0.787)
**45**	Antineoplastic (0.838) Proliferative diseases treatment (0.729) Antimetastatic (0.551)	Anti-hypercholesterolemic (0.909) Hypolipemic (0.786)	Antifungal (0.789) Anti-inflammatory (0.674) Cardiotonic (0.626)
**46**	Antineoplastic (0.665) Antimetastatic (0.626)		Antiprotozoal (Plasmodium) (0.854) Antifungal (0.642)
**47**	Antineoplastic (0.790) Proliferative diseases treatment (0.652)	Anti-hypercholesterolemic (0.963) Atherosclerosis treatment (0.742)	Antifungal (0.798) Anti-inflammatory (0.722)
**48**	Proliferative diseases treatment (0.953) Anticarcinogenic (0.922) Antineoplastic (0.911)	Anti-hypercholesterolemic (0.962) Hypolipemic (0.825) Atherosclerosis treatment (0.588)	Neuroprotector (0.976) Antimycobacterial (0.929) Antifungal (0.885)
**49**	Chemopreventive (0.960) Proliferative diseases treatment (0.945) Anticarcinogenic (0.881) Antineoplastic (0.877)	Anti-hypercholesterolemic (0.964) Hypolipemic (0.797)	Antimycobacterial (0.936) Antifungal (0.847) Anti-inflammatory (0.823) Neuroprotector (0.757)
**50**	Chemopreventive (0.953) Proliferative diseases treatment (0.944) Antineoplastic (0.858)	Anti-hypercholesterolemic (0.943) Hypolipemic (0.846) Atherosclerosis treatment (0.510)	Respiratory analeptic (0.957) Neuroprotector (0.892) Antifungal (0.873)
**51**	Proliferative diseases treatment (0.976) Chemopreventive (0.959) Antineoplastic (0.906) Dementia treatment (0.631)	Anti-hypercholesterolemic (0.976) Hypolipemic (0.735) Lipid metabolism regulator (0.707)	Respiratory analeptic (0.992) Neuroprotector (0.953) Antifungal (0.838) Antiviral (Influenza) (0.653)
**52**	Antineoplastic (0.840) Apoptosis agonist (0.581)	Hypolipemic (0.682) Cholesterol synthesis inhibitor (0.508)	Immunosuppressant (0.671) Nootropic (0.548)
**53**	Antineoplastic (0.790) Proliferative diseases treatment (0.667)	Anti-hypercholesterolemic (0.729) Cholesterol synthesis inhibitor (0.664)	Anti-osteoporotic (0.965) Anti-eczematic (0.910)
**54**	Antineoplastic (0.790) Proliferative diseases treatment (0.767)	Anti-hypercholesterolemic (0.729) Cholesterol synthesis inhibitor (0.614)	Anti-osteoporotic (0.965) Anti-eczematic (0.910)
**55**	Antineoplastic (0.948) Alzheimer’s disease treatment (0.851) Apoptosis agonist (0.696)	Anti-hypercholesterolemic (0.821) Neurodegenerative diseases treatment (0.815) Cholesterol synthesis inhibitor (0.741)	Immunomodulator (HIV) (0.893) Antioxidant (0.821)
**56**	Chemopreventive (0.918) Proliferative diseases treatment (0.874) Antineoplastic (0.845)	Anti-hypercholesterolemic (0.914) Acute neurologic disorders treatment (0.632) Atherosclerosis treatment (0.612)	Anti-inflammatory (0.870) Immunosuppressant (0.818)
**57**	Antineoplastic (0.784) Apoptosis agonist (0.654) Prostate disorders treatment (0.625)		Antiprotozoal (Plasmodium) (0.766) Anti-inflammatory (0.752) Immunosuppressant (0.705)
**58**	Antineoplastic (0.795) Apoptosis agonist (0.711) Prostate disorders treatment (0.633)		Antiprotozoal (Plasmodium) (0.754) Anti-inflammatory (0.732) Immunosuppressant (0.712)

* Only activities with Pa > 0.5 are shown.

**Table 4 marinedrugs-18-00613-t004:** Probable biological activities of steroids derived from soft coral species estimated by PASS.

No.	Antitumor and Related Activity, (Pa) *	Lipid Metabolism Regulators, (Pa) *	Additional biological activity, (Pa) *
**59**	Antineoplastic (0.886) Apoptosis agonist (0.648)	Anti-hypercholesterolemic (0.819) Cholesterol synthesis inhibitor (0.570)	Respiratory analeptic (0.888) Immunomodulator (HIV) (0.850)
**60**	Antineoplastic (0.843) Apoptosis agonist (0.780) Dementia treatment (0.528)	Neuroprotector (0.611) Nootropic (0.587) Hypolipemic (0.536)	Respiratory analeptic (0.835)
**61**	Antineoplastic (0.892) Apoptosis agonist (0.639)		Anti-inflammatory (0.599)
**62**	Antineoplastic (skin cancer) (0.650) Antineoplastic (0.650)		Anti-inflammatory (0.608) Immunomodulator (HIV) (0.568)
**63**	Antineoplastic (0.788) Apoptosis agonist (0.628)	Hypolipemic (0.738) Cholesterol synthesis inhibitor (0.660)	Antifungal (0.699) Anti-inflammatory (0.623)
**64**	Antineoplastic (0.824) Apoptosis agonist (0.634)	Hypolipemic (0.707) Cholesterol synthesis inhibitor (0.533)	Antifungal (0.777)
**65**	Antineoplastic (0.841) Proliferative diseases treatment (0.732) Apoptosis agonist (0.713)	Anti-hypercholesterolemic (0.848) Hypolipemic (0.762) Cholesterol synthesis inhibitor (0.615)	Respiratory analeptic (0.947) Immunosuppressant (0.834)
**66**	Antineoplastic (0.761) Proliferative diseases treatment (0.575)	Anti-hypercholesterolemic (0.824) Atherosclerosis treatment (0.636)	Respiratory analeptic (0.833)
**67**	Chemopreventive (0.882) Antineoplastic (0.830) Proliferative diseases treatment (0.738)	Anti-hypercholesterolemic (0.884) Atherosclerosis treatment (0.679)	Anti-inflammatory (0.860) Respiratory analeptic (0.848)
**68**	Antineoplastic (0.744)		Anti-inflammatory (0.691)
**69**	Antineoplastic (0.875) Proliferative diseases treatment (0.856)	Anti-hypercholesterolemic (0.918) Atherosclerosis treatment (0.618)	Anti-inflammatory (0.903) Nootropic (0.654)
**70**	Antineoplastic (0.822) Prostate disorders treatment (0.796)	Anti-hypercholesterolemic (0.634)	Erythropoiesis stimulant (0.858) Immunomodulator (HIV) (0.858)
**71**	Apoptosis agonist (0.964) Chemopreventive (0.906) Antineoplastic (0.854)	Atherosclerosis treatment (0.865) Lipoprotein disorders treatment (0.761) Anti-hypercholesterolemic (0.694)	Antiprotozoal (Plasmodium) (0.876) Respiratory analeptic (0.766)
**72**	Apoptosis agonist (0.982) Chemopreventive (0.945) Antineoplastic (0.920) Antiparkinsonian, rigidity relieving (0.730)	Atherosclerosis treatment (0.909) Hypolipemic (0.831) Lipoprotein disorders treatment (0.816) Anti-hypercholesterolemic (0.799)	Antiprotozoal (Plasmodium) (0.845)
**73**	Apoptosis agonist (0.970) Chemopreventive (0.929) Antineoplastic (0.883) Antiparkinsonian, rigidity relieving (0.614)	Atherosclerosis treatment (0.904) Anti-hypercholesterolemic (0.856) Hypolipemic (0.838) Lipoprotein disorders treatment (0.733)	Antiprotozoal (Plasmodium) (0.828)
**74**	Antineoplastic (0.891) Proliferative diseases treatment (0.600)	Hypolipemic (0.683)	Anti-inflammatory (0.816) Antiprotozoal (Plasmodium) (0.719)
**75**	Antineoplastic (0.932) Proliferative diseases treatment (0.611)	Anti-hypercholesterolemic (0.612) Autoimmune disorders treatment (0.587)	Anti-inflammatory (0.886)
**76**	Antineoplastic (0.933) Proliferative diseases treatment (0.623)		Anti-inflammatory (0.902) Respiratory analeptic (0.823)
**77**	Antineoplastic (0.905) Chemopreventive (0.617)	Autoimmune disorders treatment (0.640)	Anti-inflammatory (0.863) Antiprotozoal (Plasmodium) (0.653)
**78**	Apoptosis agonist (0.854) Antineoplastic (0.787)	Anti-hypercholesterolemic (0.769) Atherosclerosis treatment (0.635)	Respiratory analeptic (0.941)
**79**	Antineoplastic (0.887) Apoptosis agonist (0.880) Chemopreventive (0.605)	Anti-hypercholesterolemic (0.808) Hypolipemic (0.787) Atherosclerosis treatment (0.689)	Anti-inflammatory (0.682)
**80**	Antineoplastic (0.799) Apoptosis agonist (0.701)	Atherosclerosis treatment (0.611) Cholesterol synthesis inhibitor (0.589)	Antifungal (0.751) Antithrombotic (0.558)
**81**	Antineoplastic (0.877) Apoptosis agonist (0.838)	Anti-hypercholesterolemic (0.638) Cholesterol synthesis inhibitor (0.628)	Respiratory analeptic (0.945)
**82**	Antineoplastic (0.857)		Antifungal (0.786)
**83**	Antineoplastic (0.753) Alzheimer’s disease treatment (0.674)		Anti-inflammatory (0.733) Immunosuppressant (0.723)
**84**	Antineoplastic (0.852) Apoptosis agonist (0.760)		Respiratory analeptic (0.871) Immunosuppressant (0.781)

* Only activities with Pa > 0.5 are shown.

**Table 5 marinedrugs-18-00613-t005:** Probable biological activities of steroids derived from soft coral species estimated by PASS.

No.	Antitumor and Related Activity, (Pa) *	Lipid Metabolism Regulators, (Pa) *	Additional Biological Activity, (Pa) *
**85**	Apoptosis agonist (0.902) Antineoplastic (0.878) Antiparkinsonian, rigidity relieving (0.507)	Anti-hypercholesterolemic (0.894) Hypolipemic (0.785) Atherosclerosis treatment (0.650)	Anti-inflammatory (0.829) Immunosuppressant (0.799)
**86**	Antineoplastic (0.891)	Anti-hypercholesterolemic (0.841) Neuroprotector (0.727)	Angiogenesis inhibitor (0.910) Respiratory analeptic (0.848)
**87**	Antineoplastic (0.864)	Anti-hypercholesterolemic (0.836) Neuroprotector (0.806)	Anesthetic general (0.910) Angiogenesis inhibitor (0.895)
**88**	Antineoplastic (0.868)	Anti-hypercholesterolemic (0.623) Hypolipemic (0.521)	Angiogenesis inhibitor (0.765) Anti-asthmatic (0.630)
**89**	Chemopreventive (0.882) Antineoplastic (0.830) Apoptosis agonist (0.668)	Anti-hypercholesterolemic (0.884) Atherosclerosis treatment (0.679) Hypolipemic (0.623)	Anti-inflammatory (0.860) Respiratory analeptic (0.848) Anti-asthmatic (0.542)
**90**	Chemopreventive (0.761) Antineoplastic (0.761)	Anti-hypercholesterolemic (0.824) Atherosclerosis treatment (0.636)	Respiratory analeptic (0.833) Immunosuppressant (0.740)
**91**	Apoptosis agonist (0.767) Antineoplastic (0.597)	Hypolipemic (0.520) Atherosclerosis treatment (0.500)	Antiprotozoal (0.723) Immunosuppressant (0.534)
**92**	Antineoplastic (0.871) Apoptosis agonist (0.859)	Anti-hypercholesterolemic (0.832) Atherosclerosis treatment (0.687)	Nootropic (0.744) Immunosuppressant (0.733)
**93**	Antineoplastic (0.812) Apoptosis agonist (0.737) Chemopreventive (0.650)	Hypolipemic (0.764) Anti-hypercholesterolemic (0.686) Atherosclerosis treatment (0.575)	Immunosuppressant (0.652) Nootropic (0.602)
**94**	Antineoplastic (0.935) Apoptosis agonist (0.833)	Anti-hypercholesterolemic (0.789)	Immunomodulator (HIV) (0.958) Respiratory analeptic (0.923)
**95**	Antineoplastic (0.860) Apoptosis agonist (0.844)	Hypolipemic (0.748) Anti-hypercholesterolemic (0.536)	Immunosuppressant (0.773) Allergic conjunctivitis treatment (0.608)
**96**	Antineoplastic (0.773)	Hypolipemic (0.595)	Anti-eczematic (0.864)
**97**	Antineoplastic (0.776)	Hypolipemic (0.559)	Anti-eczematic (0.856)
**98**	Antineoplastic (0.809)	Anti-hypercholesterolemic (0.672)	Respiratory analeptic (0.935)

* Only activities with Pa > 0.5 are shown.

**Table 6 marinedrugs-18-00613-t006:** Probable biological activities of steroids derived from soft coral species estimated by PASS.

No.	Antitumor and Related Activity, (Pa) *	Lipid Metabolism Regulators, (Pa) *	Additional Biological Activity, (Pa) *
**99**	Antineoplastic (0.893) Apoptosis agonist (0.791)	Hypolipemic (0.736) Atherosclerosis treatment (0.717)	Respiratory analeptic (0.842) Antifungal (0.821)
**100**	Antineoplastic (0.895) Apoptosis agonist (0.793)	Hypolipemic (0.738) Atherosclerosis treatment (0.712)	Respiratory analeptic (0.849) Antifungal (0.825)
**101**	Antineoplastic (0.887) Apoptosis agonist (0.764) Proliferative diseases treatment (0.712) Dementia treatment (0.629)	Anti-hypercholesterolemic (0.906) Neuroprotector (0.892) Hypolipemic (0.733) Autoimmune disorders treatment (0.675)	Respiratory analeptic (0.934) Immunomodulator (HIV) (0.925) Anesthetic general (0.786) Antiviral (Influenza) (0.664)
**102**	Antineoplastic (0.870) Prostate cancer treatment (0.786) Apoptosis agonist (0.642)	Atherosclerosis treatment (0.879) Hypolipemic (0.823) Cholesterol synthesis inhibitor (0.736)	Immunomodulator (HIV) (0.939) Respiratory analeptic (0.932)
**103**	Proliferative diseases treatment (0.834) Antineoplastic (0.805)	Anti-hypercholesterolemic (0.918) Hypolipemic (0.818) Atherosclerosis treatment (0.655) Cholesterol synthesis inhibitor (0.650)	Analeptic (0.864) Antiviral (Influenza) (0.789) Antiprotozoal (0.691)
**104**	Proliferative diseases treatment (0.886) Antineoplastic (0.886) Anticarcinogenic (0.832) Dementia treatment (0.752) Alzheimer’s disease treatment (0.700)	Neurodegenerative diseases treatment (0.566)	Anti-inflammatory (0.849) Antiprotozoal (0.828)
**105**	Apoptosis agonist (0.931) Antineoplastic (0.892) Chemopreventive (0.793)	Hypolipemic (0.807) Atherosclerosis treatment (0.607) Autoimmune disorders treatment (0.531)	Nootropic (0.895) Anti-inflammatory (0.854) Antiprotozoal (Plasmodium) (0.633)
**106**	Antineoplastic (0.760)		Antiprotozoal (0.593)
**107**	Antineoplastic (0.881) Proliferative diseases treatment (0.823) Dementia treatment (0.795) Alzheimer’s disease treatment (0.688)	Neurodegenerative diseases treatment (0.562) Anti-hypercholesterolemic (0.539) Andropause treatment (0.527)	Antiprotozoal (0.862) Respiratory analeptic (0.846) Nootropic (0.671)
**108**	Antineoplastic (0.743) Prostate disorders treatment (0.719)	Anti-hypercholesterolemic (0.613)	Immunomodulator (HIV) (0.776)
**109**	Antineoplastic (0.871) Apoptosis agonist (0.649) Prostate disorders treatment (0.634)	Neuroprotector (0.981) Anti-hypercholesterolemic (0.946) Acute neurologic disorders treatment (0.749)	Respiratory analeptic (0.941) Vasoprotector (0.883) Antiprotozoal (Leishmania) (0.870)
**110**	Antineoplastic (0.860) Chemopreventive (0.812) Proliferative diseases treatment (0.694)	Anti-hypercholesterolemic (0.934) Hypolipemic (0.852) Cholesterol synthesis inhibitor (0.746) Atherosclerosis treatment (0.645)	Nootropic (0.810) Respiratory analeptic (0.785) Immunomodulator (HIV) (0.730)
**111**	Chemopreventive (0.873) Antineoplastic (0.853) Proliferative diseases treatment (0.814)	Anti-hypercholesterolemic (0.809) Neuroprotector (0.731) Hypolipemic (0.612)	Respiratory analeptic (0.941) Hepatoprotectant (0.891) Antithrombotic (0.618)

* Only activities with Pa > 0.5 are shown.

**Table 7 marinedrugs-18-00613-t007:** Probable biological activities of steroids derived from soft coral species estimated by PASS.

No.	Antitumor and Related Activity, (Pa) *	Lipid Metabolism Regulators, (Pa) *	Additional Biological Activity, (Pa) *
**112**	Antineoplastic (0.864) Apoptosis agonist (0.850) Prostate disorders treatment (0.705) Dementia treatment (0.629)	Neuroprotector (0.786) Anti-hypercholesterolemic (0.774) Cholesterol synthesis inhibitor (0.524)	Immunomodulator (HIV) (0.895) Erythropoiesis stimulant (0.894) Anesthetic general (0.865)
**113**	Antineoplastic (0.848) Apoptosis agonist (0.826) Proliferative diseases treatment (0.608)	Anti-hypercholesterolemic (0.811) Lipid metabolism regulator (0.673)	Respiratory analeptic (0.911) Neuroprotector (0.825) Immunomodulator (HIV) (0.820)
**114**	Proliferative diseases treatment (0.923) Antineoplastic (0.889) Chemopreventive (0.853) Apoptosis agonist (0.741) Dementia treatment (0.673)	Neuroprotector (0.950) Anti-hypercholesterolemic (0.821) Spasmolytic (0.684) Hypolipemic (0.621)	Respiratory analeptic (0.981) Anesthetic general (0.918) Cardiotonic (0.813) Antithrombotic (0.567)
**115**	Chemopreventive (0.942) Antineoplastic (0.901) Apoptosis agonist (0.837) Proliferative diseases treatment (0.788) Dementia treatment (0.665)	Neuroprotector (0.971) Anti-hypercholesterolemic (0.877)	Respiratory analeptic (0.987) Anesthetic general (0.938) Antiprotozoal (Leishmania) (0.927) Cardiotonic (0.791) Antithrombotic (0.557)
**116**	Chemopreventive (0.918) Antineoplastic (0.893) Proliferative diseases treatment (0.890) Dementia treatment (0.738)	Neuroprotector (0.983) Anti-hypercholesterolemic (0.919) Acute neurologic disorders treatment (0.636) Hypolipemic (0.626)	Respiratory analeptic (0.989) Anesthetic general (0.949) Antiprotozoal (Leishmania) (0.936)
**117**	Antineoplastic (0.757) Prostate disorders treatment (0.678) Proliferative diseases treatment (0.599)	Neuroprotector (0.852) Anti-hypercholesterolemic (0.828) Lipid metabolism regulator (0.732)	Respiratory analeptic (0.853) Erythropoiesis stimulant (0.851) Immunomodulator (HIV) (0.829)
**118**	Antineoplastic (0.869) Apoptosis agonist (0.775) Alzheimer’s disease treatment (0.571)		Anti-eczematic (0.924) Immunosuppressant (0.790)
**119**	Antineoplastic (0.900) Prostate disorders treatment (0.741)		Anti-eczematic (0.842) Erythropoiesis stimulant (0.827)
**120**	Antineoplastic (0.869) Apoptosis agonist (0.775) Alzheimer’s disease treatment (0.671)		Anti-eczematic (0.924) Immunosuppressant (0.790) Anti-asthmatic (0.644)
**121**	Antineoplastic (0.863) Apoptosis agonist (0.724)		Anti-inflammatory (0.834) Immunosuppressant (0.798)
**122**	Antineoplastic (0.747) Proliferative diseases treatment (0.641)	Lipid metabolism regulator (0.507)	Anti-inflammatory (0.788) Immunosuppressant (0.735)
**123**	Antineoplastic (0.762) Chemopreventive (0.643) Proliferative diseases treatment (0.619)	Anti-hypercholesterolemic (0.671) Cholesterol synthesis inhibitor (0.543)	Anti-eczematic (0.868) Immunosuppressant (0.755) Anesthetic general (0.549)
**124**	Antineoplastic (0.882) Prostate disorders treatment (0.648)		Anti-inflammatory (0.904) Antiprotozoal (0.818)
**125**	Antineoplastic (0.877) Apoptosis agonist (0.773) Alzheimer’s disease treatment (0.721)	Neurodegenerative diseases treatment (0.718)	Spasmolytic, urinary (0.959) Anti-eczematic (0.924) Immunosuppressant (0.794)
**126**	Antineoplastic (0.870) Apoptosis agonist (0.683)		Immunosuppressant (0.788) Growth stimulant (0.537)
**127**	Antineoplastic (skin cancer) (0.650) Antineoplastic (0.650)		Antibacterial (0.639) Immunomodulator (HIV) (0.568)
**128**	Antineoplastic (0.813) Apoptosis agonist (0.685) Chemopreventive (0.542)	Hypolipemic (0.781) Anti-hypercholesterolemic (0.594) Cholesterol synthesis inhibitor (0.552)	Nootropic (0.610) Immunosuppressant (0.601)
**129**	Apoptosis agonist (0.844) Antineoplastic (0.796) Chemopreventive (0.631)	Hypolipemic (0.575) Anti-hypercholesterolemic (0.551) Cholesterol synthesis inhibitor (0.507)	Anesthetic (0.689) Anti-inflammatory (0.581)
**130**	Antineoplastic (0.858) Apoptosis agonist (0.853) Chemopreventive (0.840) Prostate disorders treatment (0.539)	Anti-hypercholesterolemic (0.894) Hypolipemic (0.762) Lipid metabolism regulator (0.615) Neuroprotector (0.611)	Anti-psoriatic (0.762) Anti-eczematic (0.750) Nootropic (0.647) Anesthetic (0.602)
**131**	Antineoplastic (0.906) Chemopreventive (0.826) Apoptosis agonist (0.785)	Anti-hypercholesterolemic (0.711) Lipid metabolism regulator (0.695) Hypolipemic (0.599)	Immunosuppressant (0.735) Anti-inflammatory (0.724) Urolithiasis treatment (0.722)
**132**	Antineoplastic (0.901) Prostate disorders treatment (0.670) Apoptosis agonist (0.627)	Anti-hypercholesterolemic (0.770) Neuroprotector (0.700) Hypolipemic (0.597)	Angiogenesis inhibitor (0.920) Respiratory analeptic (0.901) Anesthetic general (0.807)

*Only activities with Pa > 0.5 are shown.

**Table 8 marinedrugs-18-00613-t008:** Probable biological activities of steroids derived from soft coral species estimated by PASS.

No.	Antitumor and Related Activity, (Pa) *	Lipid Metabolism Regulators, (Pa) *	Additional Biological Activity, (Pa) *
**133**	Antineoplastic (0.819) Prostate disorders treatment (0.701)		Respiratory analeptic (0.793) Immunomodulator (HIV) (0.743)
**134**	Antineoplastic (0.888) Proliferative diseases treatment (0.699) Anticarcinogenic (0.618)		Ankylosing spondylitis treatment (0.839) Antiprotozoal (0.773) Antiprotozoal (Plasmodium) (0.697)
**135**	Antineoplastic (0.900)	Neurodegenerative diseases treatment (0.519)	Anti-inflammatory (0.861)
**136**	Antineoplastic (0.871) Apoptosis agonist (0.585)		Antiprotozoal (0.763) Antiprotozoal (Plasmodium) (0.683)
**137**	Antineoplastic (0.880) Proliferative diseases treatment (0.607)		Antiprotozoal (0.821) Antiprotozoal (Plasmodium) (0.700)
**138**	Antineoplastic (0.856) Prostate disorders treatment (0.651) Proliferative diseases treatment (0.586)	Anti-hypercholesterolemic (0.813) Neuroprotector (0.734)	Anti-inflammatory (0.929) Immunomodulator (HIV) (0.868) Anesthetic general (0.813)
**139**	Antineoplastic (0.842)	Neuroprotector (0.558)	Immunomodulator (HIV) (0.751)
**140**	Antineoplastic (0.780) Apoptosis agonist (0.562)	Neuroprotector (0.722) Anti-hypercholesterolemic (0.609)	Immunomodulator (HIV) (0.861)
**141**	Antineoplastic (0.862) Chemoprotective (0.694)	Hypolipemic (0.554) Cholesterol synthesis inhibitor (0.509)	Immunosuppressant (0.753) Antiprotozoal (Plasmodium) (0.658)
**142**	Chemopreventive (0.989) Proliferative diseases treatment (0.969) Antineoplastic (0.874) Alzheimer’s disease treatment (0.570)	Anti-hypercholesterolemic (0.977) Neuroprotector (0.895) Atherosclerosis treatment (0.601) Neurodegenerative diseases treatment (0.590)	Hepatoprotectant (0.986) Respiratory analeptic (0.978) Antimycobacterial (0.939) Antiprotozoal (Leishmania) (0.772)
**143**	Antineoplastic (0.749)		Anti-eczematic (0.729) Dermatologic (0.651) Anti-psoriatic (0.570)
**144**	Antineoplastic (0.796) Apoptosis agonist (0.750)	Hypolipemic (0.660) Cholesterol synthesis inhibitor (0.510)	Hepatoprotectant (0.748) Anti-eczematic (0.739)
**145**	Antineoplastic (0.697) Antineoplastic (bladder cancer) (0.568)		Antibacterial (0.688) Antifungal (0.620)
**146**	Antineoplastic (0.735) Prostate disorders treatment (0.696)	Neuroprotector (0.580)	Immunomodulator (HIV) (0.817) Anti-eczematic (0.808)
**147**	Antineoplastic (0.828) Proliferative diseases treatment (0.699)	Neuroprotector (0.640) Acute neurologic disorders treatment (0.626)	Anti-inflammatory (0.920) Respiratory analeptic (0.838)
**148**	Antineoplastic (0.837) Apoptosis agonist (0.607)	Neuroprotector (0.684) Autoimmune disorders treatment (0.611)	Anti-inflammatory (0.889) Antiprotozoal (Leishmania) (0.571)
**149**	Antineoplastic (0.851) Proliferative diseases treatment (0.687)	Neuroprotector (0.771) Acute neurologic disorders treatment (0.600)	Anti-inflammatory (0.923) Immunomodulator (HIV) (0.859)
**150**	Antineoplastic (0.849) Proliferative diseases treatment (0.709)	Neuroprotector (0.736) Anti-hypercholesterolemic (0.634) Acute neurologic disorders treatment (0.615)	Anti-inflammatory (0.910) Antiprotozoal (Leishmania) (0.584)
**151**	Antineoplastic (0.818) Apoptosis agonist (0.571)		Respiratory analeptic (0.858) Cardiotonic (0.612)
**152**	Antineoplastic (0.832) Proliferative diseases treatment (0.578)	Anti-hypercholesterolemic (0.789)	Respiratory analeptic (0.894) Immunomodulator (HIV) (0.857)
**153**	Antineoplastic (0.837) Proliferative diseases treatment (0.598)	Anti-hypercholesterolemic (0.789)	Respiratory analeptic (0.897) Immunomodulator (HIV) (0.858)
**154**	Apoptosis agonist (0.862) Antineoplastic (0.846) Proliferative diseases treatment (0.623)	Anti-hypercholesterolemic (0.911) Hypolipemic (0.751) Atherosclerosis treatment (0.611)	Antidiabetic (type 2) (0.617) Antifungal (0.584)
**155**	Antineoplastic (0.738) Proliferative diseases treatment (0.647)	Anti-hypercholesterolemic (0.845) Cholesterol synthesis inhibitor (0.562)	Respiratory analeptic (0.911) Myocardial infarction treatment (0.906)
**156**	Antineoplastic (0.737)	Anti-hypercholesterolemic (0.538)	Myocardial infarction treatment (0.889) Respiratory analeptic (0.813)

* Only activities with Pa > 0.5 are shown.

**Table 9 marinedrugs-18-00613-t009:** Probable biological activities of steroids derived from soft coral species estimated by PASS.

No.	Antitumor and Related Activity, (Pa) *	Lipid Metabolism Regulators, (Pa) *	Additional Biological Activity, (Pa) *
**157**	Antineoplastic (0.749)	Anti-hypercholesterolemic (0.825)	Anti-inflammatory (0.881)
**158**	Antineoplastic (0.701) Prostate disorders treatment (0.629)	Anti-hypercholesterolemic (0.929) Hypolipemic (0.645)	Anti-seborrheic (0.868) Radioprotector (0.801)
**159**	Apoptosis agonist (0.791) Antineoplastic (0.787)	Hypolipemic (0.680)	Anti-psoriatic (0.795) Immunosuppressant (0.726)
**160**	Antineoplastic (0.913) Apoptosis agonist (0.862) Proliferative diseases treatment (0.687)	Atherosclerosis treatment (0.628) Anti-hypercholesterolemic (0.587)	Anti-eczematic (0.844) Anti-psoriatic (0.820)
**161**	Antineoplastic (0.860) Apoptosis agonist (0.851) Proliferative diseases treatment (0.700)	Anti-hypercholesterolemic (0.789) Atherosclerosis treatment (0.724) Cholesterol synthesis inhibitor (0.617)	Anti-osteoporotic (0.913) Nootropic (0.870)
**162**	Antineoplastic (0.872) Apoptosis agonist (0.811) Proliferative diseases treatment (0.786)	Anti-hypercholesterolemic (0.943) Hypolipemic (0.730) Atherosclerosis treatment (0.648)	Anesthetic general (0.917) Respiratory analeptic (0.869) Nootropic (0.698)
**163**	Antineoplastic (0.937) Apoptosis agonist (0.906) Proliferative diseases treatment (0.886) Antiparkinsonian, rigidity relieving (0.729) Multiple sclerosis treatment (0.565)		Anti-eczematic (0.973) Anti-psoriatic (0.963) Anti-osteoporotic (0.961) Hyperparathyroidism treatment (0.842) Hypercalcemia treatment (0.509)
**164**	Antineoplastic (0.706) Proliferative diseases treatment (0.608)	Anti-hypercholesterolemic (0.882) Atherosclerosis treatment (0.723) Cholesterol synthesis inhibitor (0.632)	Hepatic disorders treatment (0.824) Anti-osteoporotic (0.665)
**165**	Antineoplastic (0.750)	Anti-hypercholesterolemic (0.791) Atherosclerosis treatment (0.700) Acute neurologic disorders treatment (0.635)	Hepatic disorders treatment (0.816) Anesthetic general (0.774) Anti-osteoporotic (0.663)
**166**	Antineoplastic (0.694) Apoptosis agonist (0.578) Proliferative diseases treatment (0.542)	Anti-hypercholesterolemic (0.852) Hypolipemic (0.788) Atherosclerosis treatment (0.744)	Anti-eczematic (0.870) Hepatic disorders treatment (0.817) Anti-osteoporotic (0.678)
**167**	Antineoplastic (0.789) Apoptosis agonist (0.726) Proliferative diseases treatment (0.608)	Anti-hypercholesterolemic (0.891) Atherosclerosis treatment (0.630) Acute neurologic disorders treatment (0.560)	Anti-eczematic (0.894) Anti-psoriatic (0.777) Anti-osteoporotic (0.736)
**168**	Proliferative diseases treatment (0.947) Dementia treatment (0.657)	Neuroprotector (0.981) Anti-hypercholesterolemic (0.971) Hypolipemic (0.778)	Respiratory analeptic (0.987) Hepatoprotectant (0.945) Anesthetic general (0.903)
**169**	Proliferative diseases treatment (0.920) Antineoplastic (0.841) Dementia treatment (0.661)	Anti-hypercholesterolemic (0.962) Neuroprotector (0.952) Acute neurologic disorders treatment (0.783)	Respiratory analeptic (0.991) Anesthetic general (0.922) Spasmolytic (0.719)
**170**	Antineoplastic (0.793) Proliferative diseases treatment (0.605)		Anesthetic general (0.815) Cardiotonic (0.770)
**171**	Antineoplastic (0.872) Alzheimer’s disease treatment (0.634) Proliferative diseases treatment (0.625)	Neuroprotector (0.732) Hypolipemic (0.592) Acute neurologic disorders treatment (0.557)	Anti-inflammatory (0.923) Nootropic (0.672)
**172**	Antineoplastic (0.871) Alzheimer’s disease treatment (0.681) Dementia treatment (0.576)	Neuroprotector (0.762) Anti-hypercholesterolemic (0.633) Neurodegenerative diseases treatment (0.618)	Respiratory analeptic (0.894) Anesthetic general (0.890)
**173**	Proliferative diseases treatment (0.922) Antineoplastic (0.842)	Anti-hypercholesterolemic (0.859) Neuroprotector (0.706)	Respiratory analeptic (0.962) Anti-ischemic, cerebral (0.864)
**174**	Proliferative diseases treatment (0.935) Antineoplastic (0.837) Dementia treatment (0.616)	Anti-hypercholesterolemic (0.927) Neuroprotector (0.773) Hypolipemic (0.584)	Anti-ischemic, cerebral (0.944) Respiratory analeptic (0.940) Antithrombotic (0.720)
**175**	Antineoplastic (0.793)	Anti-hypercholesterolemic (0.954) Neuroprotector (0.754) Hypolipemic (0.704)	Respiratory analeptic (0.979) Anesthetic general (0.927) Anti-ischemic, cerebral (0.679)
**176**	Antineoplastic (0.893) Proliferative diseases treatment (0.768)	Anti-hypercholesterolemic (0.899) Atherosclerosis treatment (0.754)	Respiratory analeptic (0.979) Immunomodulator (HIV) (0.912)
**177**	Antineoplastic (0.812) Dementia treatment (0.505)	Neuroprotector (0.938) Anti-hypercholesterolemic (0.912) Atherosclerosis treatment (0.548)	Respiratory analeptic (0.984) Immunomodulator (HIV) (0.933) Anesthetic general (0.917)

* Only activities with Pa > 0.5 are shown.

**Table 10 marinedrugs-18-00613-t010:** Probable biological activities of steroids derived from soft coral species estimated by PASS.

No.	Antitumor and Related Activity, (Pa) *	Lipid Metabolism Regulators, (Pa) *	Additional Biological Activity, (Pa) *
**178**	Antineoplastic (0.886) Apoptosis agonist (0.829) Proliferative diseases treatment (0.710)	Anti-hypercholesterolemic (0.716) Hypolipemic (0.596) Atherosclerosis treatment (0.588)	Respiratory analeptic (0.966) Immunosuppressant (0.837) Anesthetic general (0.775)
**179**	Antineoplastic (0.901) Proliferative diseases treatment (0.832)	Anti-hypercholesterolemic (0.839) Atherosclerosis treatment (0.597)	Respiratory analeptic (0.979) Anesthetic general (0.841)
**180**	Antineoplastic (0.897) Apoptosis agonist (0.858) Proliferative diseases treatment (0.778)	Anti-hypercholesterolemic (0.865) Atherosclerosis treatment (0.639)	Respiratory analeptic (0.974) Anesthetic (0.904) Anesthetic general (0.731)
**181**	Antineoplastic (0.842) Apoptosis agonist (0.779) Proliferative diseases treatment (0.711)	Anti-hypercholesterolemic (0.658) Hypolipemic (0.633)	Respiratory analeptic (0.941) Anesthetic (0.842) Cardiotonic (0.730)
**182**	Apoptosis agonist (0.828) Antineoplastic (0.827) Proliferative diseases treatment (0.536)	Hypolipemic (0.636) Atherosclerosis treatment (0.630) Cholesterol synthesis inhibitor (0.519)	Anesthetic (0.902) Antithrombotic (0.573) Spasmolytic (0.538)
**183**	Apoptosis agonist (0.898) Antineoplastic (0.852) Proliferative diseases treatment (0.519)	Hypolipemic (0.660) Atherosclerosis treatment (0.618) Lipid metabolism regulator (0.535)	Anesthetic (0.869) Immunosuppressant (0.798) Anti-inflammatory (0.792)
**184**	Antineoplastic (0.881) Apoptosis agonist (0.797) Proliferative diseases treatment (0.611)	Neuroprotector (0.603) Hypolipemic (0.568)	Respiratory analeptic (0.915) Antifungal (0.828) Antiprotozoal (0.718)
**185**	Antineoplastic (0.937) Alzheimer’s disease treatment (0.633)		Anti-inflammatory (0.939) Antiprotozoal (0.663)
**186**	Antineoplastic (0.934) Apoptosis agonist (0.792) Antimetastatic (0502)		Anti-inflammatory (0.934) Anti-asthmatic (0.645) Antiprotozoal (Plasmodium) (0.601)
**187**	Antineoplastic (0.934) Apoptosis agonist (0.792) Antimetastatic (0.502)		Anti-inflammatory (0.934) Anti-asthmatic (0.645) Antiprotozoal (Plasmodium) (0.601)

* Only activities with Pa > 0.5 are shown.

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
