# Peer review of "Chemical Diversity of Soft Coral Steroids and Their Pharmacological Activities"

_marinedrugs, 2020, doi:10.3390/md18120613_

Round 1

Reviewer 1 Report

Dear Editor!

This review describes the soft corals steroid compounds, their structures and biological activity. It also has tables with the potential biological activity of the described compounds. I think that this review will be useful both to chemists working with soft coral metabolites and biologists working on defining of the biological activity.

During the review of the manuscript, I found several mistakes:

Line 52: “Tree” should be corrected as “Three”.

Line 52: I recommend using the same names for compounds: for example, “ergosta-5,24(28)-diene-3β,4α-diol (1), 24(S),28-epoxyergost-5-ene-3β, 4α-diol 52 (2)” instead of “3β,4α-dihydroxyergosta-5,24(28)-diene (1), 24(S),28-epoxyergost-5-ene-3β, 4α-diol 52 (2)” and further through the text.

Figure 1. I recommend adding the numbering of carbon atoms in steroid 1.

Figure 1, 2, and 3. I recommend adding protons at C-5, C-8, C-9, C-14, and C-17 for all compounds as since Figure 4 and other Figures.

All Figures. I recommend using the same type of image of carbohydrate moiety of steroidal glycosides. For example, as for compound 5.

Names of all tables. Why are tables named “Anti-tumor activity…”? The tables describe the different types of activity. Perhaps they should be named “Potential biological activity…” or “Potential pharmacological activity…”

Footnote of Table 1. I recommend adding description of “Pa”.

Line 65: Compounds 7 is tetraol having 24(28) double bond. Its name should be corrected and configuration of 7-OH should be added.

Line 79: I recommend to change name of compound 16 on “24-methyl-5β-cholest-24(28)-en-1α,3β,5β-triol-6-one”.

Line 83: Compound 18 (16α-hydroxysarcosterol ) doesn't have 16-OH in Figure 1. This error should be corrected.

Line 97 and Figure 2: Compound 28 is not unsaturated analogue of 27. Compound 28 has additional 7α-OH.

Lines 102-103: I recommend changing “polyhydroxy sterol” on “polyhydroxysteroid” or “polyhydroxylated sterol”.

Line 108: It should be added stereochemistry for “11,12-epoxy-steroid (43) and 3,6,9,19-tetraol-steroid (44)”.

Line 114: It should be deleted space “3β, 11-”

Line 119: It should be corrected expression “and two cytotoxic 5,8-epidioxy-steroid (46) and (47) were found in Sinularia sp.” because compound 47 doesn't have peroxide group.

Lines 132-133: It should be corrected expression “afforded four new steroids 132 with methyl ester groups, sinubrasones A (57) and B (58)” because authors described only two compounds (57 and 58) and steroid 57 doesn't have methyl ester group.

Line 139: It should be correct expression “Cytotoxic 19-oxygenated steroids (6367) were obtained” because compound 67 doesn't have oxygen atom at C-19.

Lines 140-141: It should be correct expression “steroid with a spiro-ring A, B system named chabrolosteroid C (63) and chabrolosteroid A (64)” because compounds 63 and 64 don't have spiro-ring A, B system.

Lines 144-145: It should be corrected the names of “5α, 8α-epidioxycholest-6-en-3β-ol (71), 8α-epidioxy-24(S)-methylcholesta-6,22-dien-3β-ol (72) and 5α,8α-epidioxy-24α-ethyl-cholesta-6,22-dien-3β-ol (73)”: to delete space for 71, to add 5α, for 72 and to change stereochemistry for C-24 of 73 as for compound 72. Stereochemistry α and β is use only for steroid nucleus but not for steroid side chain. It should be added R and S stereochemistry for steroid side chain.

Line 164. I recommend changing “Polyoxygenated” on “Polyhydroxygenated”

Line 167. It should be add value of EC50 for bioactivity.

Lines 167-168: It should be corrected names of “3β,23-ergosta-5,24(28)-diene-3,23-diol (90)” because it does not match structure 90 on Figure 5.

Line 169. I recommend changing “synthesized” on “produced”.

Lines 172, 174, and 177. I recommend to change “9,11-secosterols” on “9,11-secosteroids” and “sterol” on “steroid”, respectively.

Line 180: It should be corrected name of “16,22-Epoxy-20β,23S-dihydroxycholest-1-ene-3-one (98)” because it does not epoxy cycle, it has furan cycle.

Figure 5: It should be correct structure of 103, because it has H-5 proton.

Line 193. It should be added stereochemistry of monosaccharide unit of compound 109 (D or L)

Line 195: I recommend to change name of compound 110 on “24-methylcholest-4(5),24(28)-dien-3β,6β-diol”.

Line 196: It should be corrected “south China Sea” on “South China Sea”.

Line 203: It should be corrected expression “Another two cytotoxic named sclerosteroids D (117) and E (118)” on “Another two cytotoxic compounds named sclerosteroids D (117) and E (118)” or “Another two cytotoxic steroids named sclerosteroids D (117) and E (118)”.

Lines 204-205: It should be added stereochemistry of 4-OMe of compound 119.

Figure 6: It should be corrected the structures of 112 and 113, it should be added stereochemistry of 3-OH and 3-OAc.

Line 215: It should be corrected the expression “Minabeolides-1 (125) and -5 (126) as C28 steroidal” on “minabeolides-1 (125) and 5 (126) as C-28 steroidal”.

Lines 220 and 222. I recommend to change “secosterols” on “secosteroids” and “(22R,23S,24S)-Polyoxygenated steroid” on “(22R,23S,24S)-Polyhydroxygenated steroid”, respectively.

Lines 244 and 245. It should be corrected name of “13,14-seco-22-norergosta-4,24(28)-dien-19-hydro-peroxide-3-one (139)” on “13,14-seco-22-norergosta-4,24(28)-dien-13-hydro-peroxide-3-one (139)”.

Line 250: It should be deleted space “5β, 6-epoxy” and “3β, 11-”

Line 267. I recommend changing “polyoxygenated” on “polyhydroxygenated”.

Figure 8: It should be corrected structure of 161, it should be added two hydroxyl groups 1α-OH and 25-OH .

Line 279. I recommend changing “saponins” on “glycosides”.

Line 288. It should be specify name of sea “South Sea Korea”.

Lines 290-292: It should be correcedt names of compounds “27-O-[β-D-arabino-pyranosyloxy]-20β22α-dihydroxy-cholest-4-ene-3-one, named muricellasteroid D (172), and 22α-O-acetyl-2β-O-methylene-[4βÒ-hydroxy-phenly]-cholest-4-ene-3-one (173)”, because stereochemistry α and β is use only for steroid nucleus but not for steroid side chain. It should be added R and S stereochemistry for steroid side chain.

Figure 8: It should be corrected structures of 173 and 175: to delete stereochemistry of C-25 for 173, to add stereochemistry of 3OH for 175.

Line 297: It should be corrected name of compounds “pregna-5-ene-3β,20α,21-triol (175)”. It should be R and S stereochemistry for steroid side chain.

All texts: When listing compounds or other things, you should add comma before “and” if more than two items are listed.

Finally, the publication can be accepted with minor revision.

Author Response

All authors of this article would like to thank Reviewer 1 for helping us improve our article.
We send our response to each comment. The fixes are marked in yellow.
the answer is marked in green. The chemical structures are all checked and rewritten as recommended.

Reviewer 2 Report

In this review the chemical diversity of steroids produced by soft corals was reported. Moreover, it summarizes the pharmacological activities of these molecules, paying particular attention to anticancer activities. In general, the topic is very interesting and it is really impressive that nearly 200 structures have been reported, but there are a number of issues that need attention

1) It is clear that this review is very interesting because “this review is probably the first and exclusive to present the known as well as the potential pharmacological activities of 200 marine steroids”, as you write in the abstract (lines 23-25). I believe that the review lists very well the predicted activities with tables and the paragraph 4; on the contrary, you only added “known” activities demonstrated by in vitro experiments for a few compounds. Therefore, I think you should add more information on in vitro bioactivities (e.g. the activity found and EC50 values, if there are in the literature) for the compounds you mentioned throughout the review. This information is mainly missing in paragraph 2.

2) To make the tables more complete, a column could be added to the tables indicating the source (i.e. the organism) from which the compounds were extracted. In addition, you should expand the legends by better explaining what tables include. Please pay also attention to the order of the tables. Moreover, decimals in English are indicated with the dot.

3) You should add more information about the PASS software also in the “Introduction”. Otherwise it is possible to understand the tables in paragraphs 2 and 3 only after reading the paragraph 4. Perhaps you could move some introductory sentences on the PASS software from paragraph 4 to the "Introduction"

4) If already mentioned once in the text, scientific names of the organisms in the manuscript should be abbreviated. I have done as much as I can, but I think you should do a further check

5) Generally, the conclusions should not simply be a summary (like the abstract) of what is written in the review. Therefore, you should modify the final paragraph making it different from the abstract. It should be changed to 3-4 concluding paragraphs which include more considerations/discussions on the coral steroids, them known and possible applications, but also on the importance of the PASS software.

Author Response

Reviewer answer 2.

All authors of this article would like to thank Reviewer 2 for helping us improve our article.

1. Such information is in the text for those connections whose activity has been studied.

2. We cannot agree with the second point, since the tables are very overloaded. Any additions to the Tables are highly undesirable, and the names of the organisms are given in the text. We fixed the decimal fractions. All corrections are marked in yellow in the text.

3. We have added information on PASS as in the introduction.

4. We have removed the titles from the text where necessary.

5. According to the recommendations, we rewrote the conclusion.

Reviewer recommendations have been implemented.

Round 2

Reviewer 2 Report

In this review the chemical diversity of steroids produced by soft corals and their pharmacological activities were reported. In general, the topic is very interesting and it is really impressive that nearly 200 structures have been reported, and I'm sure this review will be useful for many researchers.

However, the authors responded to many of my comments satisfactorily. Unfortunately, there are some mistakes/comments in the text that have not been addressed, so I fear that the authors did not see my comments in the pdf file that I attached during the first submission.

Please, find attached again the pdf file with my first revision and consider comments that have not been seen previously.

I believe the manuscript can be considered for publication after minor revision.

Author Response

Dear Reviewer 2,

Thank you very much for carefully reading our article. Unfortunately, the previous time we could not find your recommendations on the magazine's website. All comments have been removed and are highlighted in green.

Thanks again.
